# Predicting mutational routes to new adaptive phenotypes

**Peter A Lind[1,2]\*, Eric Libby[1,3,4], Jenny Herzog[1], Paul B Rainey[1,5,6]**

[1]New Zealand Institute for Advanced Study, Massey University at Albany, Auckland, New Zealand; [2]Department of Molecular Biology, Umeå University, Umeå, Sweden; [3]Santa Fe Institute, New Mexico, United States; [4]Department of Mathematics, Umeå University, Umeå, Sweden; [5]Department of Microbial Population Biology, Max Planck Institute for Evolutionary Biology, Plön, Germany; [6]Ecole Supérieure de Physique et de Chimie Industrielles de la Ville de Paris, ESPCI Paris-Tech, CNRS UMR 8231, PSL Research University, Paris, France

**Abstract** Predicting evolutionary change poses numerous challenges. Here we take advantage of the model bacterium *Pseudomonas fluorescens* in which the genotype-to-phenotype map determining evolution of the adaptive 'wrinkly spreader' (WS) type is known. We present mathematical descriptions of three necessary regulatory pathways and use these to predict both the rate at which each mutational route is used and the expected mutational targets. To test predictions, mutation rates and targets were determined for each pathway. Unanticipated mutational hotspots caused experimental observations to depart from predictions but additional data led to refined models. A mismatch was observed between the spectra of WS-causing mutations obtained with and without selection due to low fitness of previously undetected WS-causing mutations. Our findings contribute toward the development of mechanistic models for forecasting evolution, highlight current limitations, and draw attention to challenges in predicting locus-specific mutational biases and fitness effects.

DOI: https://doi.org/10.7554/eLife.38822.001

\*For correspondence:
peter.lind@umu.se

## Introduction

Adaptation requires the realization of beneficial mutations. As self-evident as this may be, predicting the occurrence of beneficial mutations and their trajectories to improved fitness is fraught with challenges (*Lässig et al., 2017*). Nonetheless progress has been made for phenotypically diverse asexual populations subject to strong selection. Effective approaches have drawn upon densely sampled sequence data and equilibrium models of molecular evolution to predict amino acid preferences at specific loci (*Luksza and Lässig, 2014*). Predictive strategies have also been developed based on selection inferred from the shape of coalescent trees (*Neher et al., 2014*). In both instances the models are coarse-grained and sidestep specific molecular and mutational details.

There is reason to by-pass molecular details: mutation, being a stochastic process, means that for the most part details are likely to be idiosyncratic and unpredictable. But an increasing number of investigations give reason to think otherwise – that adaptive molecular evolution might follow rules (*Yampolsky and Stoltzfus, 2001*; *Pigliucci, 2010*; *Stern, 2013*; *Laland et al., 2015*). This is particularly apparent from studies of parallel molecular evolution (*Colosimo et al., 2005*; *Woods et al., 2006*; *Ostrowski et al., 2008*; *Flowers et al., 2009*; *Meyer et al., 2012*; *Tenaillon et al., 2012*; *Zhen et al., 2012*; *Herron and Doebeli, 2013*; *Galen et al., 2015*; *Bailey et al., 2017*; *Kram et al., 2017*; *Stoltzfus and McCandlish, 2017*), and particularly from studies that show parallel evolution to be attributable – at least in part – to factors other than selection (*McDonald et al., 2009*; *Lind et al., 2015*; *Bailey et al., 2017*).

**eLife digest** Predicting evolution might sound like an impossible task. The immense complexity of biological systems and their interactions with the environment has meant that many biologists have abandoned the idea as a lost cause. But despite this, evolution often repeats itself. This repeatability offers hope for being able to spot in advance how evolution will happen. To make general predictions, it is necessary to understand the mechanisms underlying evolutionary pathways, and studying microbes in the laboratory allows for real-time experiments in evolution.

One of the best studied microbes for experimental evolution is *Pseudomonas fluorescens*, which repeatedly evolves flattened wrinkled colonies instead of round smooth ones when there is limited oxygen. The underlying molecular pathways that lead to this change have been studied in detail.

Lind et al. developed mathematical models to predict how often the three most common pathways would be used and which genes were most likely to be mutated. After controlling for the effects of natural selection and refining the models to take into account mutation hotspots, Lind et al. were able to accurately predict the genes that would be targeted by mutations.

The findings suggest that biologists need not lose hope when it comes to the goal of predicting evolution. A deep understanding of the molecular mechanisms of evolutionary changes are essential to predicting the mutations that lead to adaptive change. The results are an important first step towards forecasting organisms' responses to changing conditions in the future. In the short term, this is important for medical issues, including antibiotic resistance, cancer and immune receptors. In the long term, predicting the course of evolution could be essential for survival of life on the planet.
DOI: https://doi.org/10.7554/eLife.38822.002

A standard starting position for predicting adaptive evolution recognises the importance of population genetic parameters including mutation rate, generation time, population size, selection and more recently information on the distribution of beneficial fitness effects. An often used approach appeals to 'origin-fixation' models that estimate the probability that selection will realise particular mutational trajectories based on the assumption that the rate of evolution is a function of the rate at which mutations originate, multiplied by their fixation probability (for review, see (*McCandlish and Stoltzfus, 2014*), for application, see for example (*Weinreich et al., 2006*)). Leaving aside a number of simplifying and restrictive assumptions, population genetic approaches such as those encompassed by origin-fixation models require pre-existing knowledge of (or assumptions concerning) mutation rate to a particular phenotype and selection coefficients (obtained in the absence of frequency dependent effects) and thus have predictive power only in specific situations and with the benefit of hindsight (*McCandlish and Stoltzfus, 2014*).

Looking to the future, there is hope that one day it might be possible to predict the course of evolutionary change in response to specific environmental challenges in much the same way as gene function, regulation and interactions can be predicted today based on knowledge of nucleotide sequence data. A central issue is to define the necessary information. Mechanistic understanding of the connection between genotype and phenotype combined with knowledge of the genomic bases of mutational bias offer opportunities for progress.

Mutations arise randomly with respect to utility, but genetic architecture can influence the translation of mutation into phenotypic variation: the likelihood that a given mutation generates phenotypic effects depends on the genotype-to-phenotype map (*Alberch, 1991*; *Gompel and Prud'homme, 2009*; *Stern and Orgogozo, 2009*; *Rainey et al., 2017*). Thus, the functions of gene products and their regulatory interactions provide information on likely mutational targets underpinning particular phenotypes. For example, consider a hypothetical structural determinant subject to both positive and negative regulation and whose over-expression generates a given adaptive phenotype. Assuming a uniform distribution of mutational events, mutations in the negative regulator (and not the positive activator) will be the primary cause of the adaptive phenotype. This follows from the fact that loss-of-function mutations are more common than gain-of-function mutations. Indeed, an emerging rule indicates that phenotypes determined by genetic pathways that are themselves subject to negative regulation are most likely to arise by loss-of-function mutations in negative regulatory components (*McDonald et al., 2009*; *Tenaillon et al., 2012*; *Lind et al., 2015*; *Fraebel et al., 2017*).

Mutation is not equally likely at each nucleotide of a given genome (*Lind and Andersson, 2008*; *Lynch, 2010*; *Seier et al., 2011*; *Foster et al., 2015*; *Reijns et al., 2015*; *Sankar et al., 2016*; *Stoltzfus and McCandlish, 2017*). Numerous instances of mutational bias have been reported. Prime examples are simple sequence repeats such as homopolymeric nucleotide tracts or di-, tri- and tetrameric repeats that mutate at high frequency via slipped strand mispairing (*Levinson and Gutman, 1987*). These readily identifiable sequences define contingency loci in obligate human pathogens and commensals (*Moxon et al., 1994*) and are widespread in eukaryotic genomes (*Tautz and Renz, 1984*). The behaviour of contingency loci can be further modulated by defects in components of methyl-directed mismatch repair systems (*Richardson and Stojiljkovic, 2001*; *Martin et al., 2004*; *Hammerschmidt et al., 2014*; *Heilbron et al., 2014*).

Certain palindromic structures also lead to mutational bias (*Viswanathan et al., 2000*; *Lovett, 2004*) and promote amplification events that can increase target size for additional mutations (*Roth et al., 1996*; *Kugelberg et al., 2010*; *Reams and Roth, 2015*) Transition-transversion bias (*Stoltzfus and McCandlish, 2017*) and elevated mutation rates at CpG sites (*Galen et al., 2015*) can further skew the distributions of mutational effects. Additional bias arises from the chromosomal neighbourhood of genes under selection (*Steinrueck and Guet, 2017*), the location of genes with regard to interactions with DNA replication/transcription machineries (*Sankar et al., 2016*), and environmental factors that affect not only mutation rate but also the spectra of mutational events (*Krašovec et al., 2017*; *Maharjan and Ferenci, 2017*; *Shewaramani et al., 2017*).

Beyond the genotype-to-phenotype map and mutational biases, predicting adaptive evolution requires ability to know *a priori* the fitness effects of specific mutations. At the present time there is much theoretical and empirical interest in the distribution of fitness effects (DFE) (*Eyre-Walker and Keightley, 2007*) — and particularly the DFE of beneficial mutations (*Orr, 2005*) — because of implications for predicting the rate of adaption and likelihood of parallel evolution (*de Visser and Krug, 2014*), but knowledge of the shape of the distribution is insufficient to connect specific mutations to their specific effects, or to their likelihood of occurrence. Such connections require a means of knowing the connection between mutations and their environment-specific fitness effects. This is a tall order. A starting point is to understand the relationship between all possible mutational routes to a particular phenotype and the set that are realised by selection.

Here we take a bacterial system in which the genetic pathways underpinning evolution of the adaptive 'wrinkly spreader' (WS) type are known and use this to explore the current limits on evolutionary forecasting. *Pseudomonas fluorescens* SBW25 growing in static broth microcosms rapidly depletes available oxygen establishing selective conditions that favour mutants able to colonise the air-liquid interface. The most successful mutant-class encompasses the WS types (*Ferguson et al., 2013*; *Lind et al., 2017b*). These types arise from mutational activation of diguanylate cyclases (DGCs) that cause over-production of the second messenger c-di-GMP (*Goymer et al., 2006*; *McDonald et al., 2009*), over-production of an acetylated cellulose polymer (*Spiers et al., 2002*; *Spiers et al., 2003*) and ultimately formation of a self-supporting microbial mat (*Figure 1A*).

McDonald *et al.* (*McDonald et al., 2009*) showed that each time the tape of WS evolution is re-run mutations generating the adaptive type arise in one of three DGC-encoding pathways (Wsp, Aws, or Mws) (*Figure 1A*). Subsequent work revealed that when these three pathways are eliminated from the ancestral type evolution proceeds along multiple new pathways (*Lind et al., 2015*) resulting in WS types whose fitnesses are indistinguishable from those arising via mutations in Wsp, Aws, or Mws. Parallel evolution of WS involving preferential usage of the Wsp, Aws and Mws pathways is therefore not explained by selection: repeated use of Wsp, Aws and Mws stems from the fact that these pathways are subject to negative regulation and thus, relative to pathways subject to positive regulation, or requiring promoter-activating mutations, gene fusion events, or other rare mutations, present a large mutational target (*Lind et al., 2015*).

Given repeatability of WS evolution, knowledge of the Wsp/Aws/Mws pathways, plus genetic tools for mechanistic investigation — including capacity to obtain WS mutants in the absence of selection — the WS system offers a rare opportunity to explore the extent to which knowledge of the genotype-to-phenotype map can lead mechanistic models for evolutionary forecasting. Our findings show that short-term mechanistic-level predictions of mutational pathways are possible, but also draw attention to challenges that stem from current inability to *a priori* predict locus-specific mutational biases and environment-specific fitness effects.

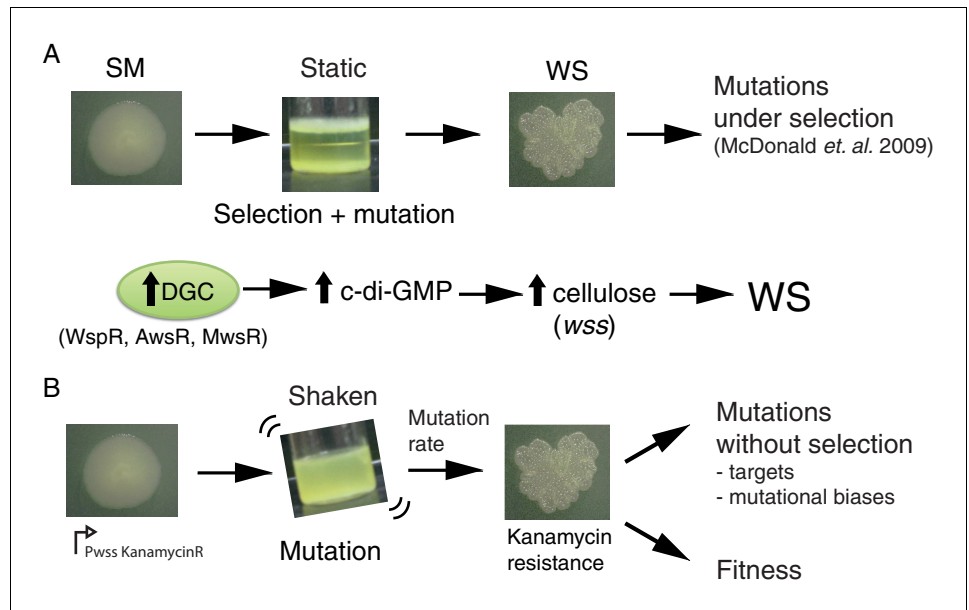

**Figure 1.** The *Pseudomonas fluorescens* SBW25 wrinkly spreader model. (**A**) Selection for access to oxygen allows wrinkly spreader (WS) mutants to invade the ancestral smooth (SM) population in static microcosms. WS mutants form a mat at the air-liquid interface through increased expression of the main structural component, cellulose, encoded by the *wss* operon. Expression of cellulose is controlled by the second messenger c-di-GMP, which is produced by diguanylate cyclases (DGCs). Mutations in the *wsp*, *aws* and *mws* operons, that activate their respective DGCs (WspR, AwsR, MwsR), are the primary mutational pathways to WS. (**B**) When a reporter construct connecting expression of the *wss* operon to resistance to kanamycin is used under shaken non-selective conditions, WS mutants can be isolated without the biasing influence of natural selection. This allows estimation of the mutation rate to WS and an unbiased spectrum of mutations defining the mutational target. Fitness can then be assayed in competition with a common reference strain.

DOI: https://doi.org/10.7554/eLife.38822.003

## Results

### Obtaining an unbiased measure of pathway-specific mutation rates to WS

Knowledge of the rate at which mutation generates WS types via each of the Wsp, Aws and Mws pathways — unbiased by the effects of selection — provides a benchmark against which the predictive power of null models can be appraised. To achieve such measures we firstly constructed a set of genotypes containing just one of the three focal pathways: PBR721 carries the Wsp pathway but is devoid of Aws and Mws, PBR713 carries the Aws pathway but is devoid of Wsp and Mws, while PBR712 harbours the Mws pathway but is devoid of Wsp and Aws. Into each of these genotypes a promoterless kanamycin resistance gene was incorporated immediately downstream of the promoter of the cellulose-encoding *wss* operon and transcriptionally fused to an otherwise unaffected *wss* operon (*Figure 1B*).

In the ancestral SM genotype the cellulose promoter is inactive in shaken King's Medium B (KB) broth (*Spiers et al., 2002*) and thus the strain is sensitive to kanamycin. When a WS-causing mutation occurs, the *wss* promoter becomes active resulting in a kanamycin-resistant WS type (*Fukami et al., 2007*; *McDonald et al., 2011*). Individual growth of this set of three genotypes in shaken KB, combined with plating to detect kanamycin-resistant mutants, makes possible a fluctuation assay (*Luria and Delbrück, 1943*; *Hall et al., 2009*) from which a direct measure of the rate at which WS mutants arise can be obtained. Importantly, because WS types are maladapted in shaken broth culture, the screen for kanamycin-resistant clones allows the pathway-specific mutation rate to WS to be obtained without the biasing effects of selection for growth at the air-liquid interface (*Figure 1B*). The results are shown in *Figure 2*.

The mutation rate was highest for the Aws pathway ($6.5 \times 10^{-9}$); approximately double that of Wsp ($3.7 \times 10^{-9}$) and an order of magnitude higher than that of the Mws pathway ($0.74 \times 10^{-9}$) (*Figure 2*). The rate at which WS mutants arose from the ancestral genotype in which the three pathways are intact ($11.2 \times 10^{-9}$) was approximately the sum of the rates for the three pathways ($11.0 \times 10^{-9}$) confirming that the Wsp, Aws and Mws pathways are the primary routes by which WS types evolve (*Lind et al., 2015*). That the Aws pathway has the greatest capacity to generate WS is surprising given the smaller target size (three genes and 2.3 kb compared to seven genes (8.4 kb) in the Wsp pathway).

## The genotype-to-phenotype map underpinning WS evolution

Much is known about the function and interactions among components of each of the three focal pathways. This knowledge allows development of models that capture the dynamic nature of each pathway and thus allow predictions as to the likelihood that evolution will precede via each of the three mutational routes. An unresolved issue is the extent to which these models match experimental findings. Following a brief description of each pathway we describe the models.

The 8.4 kb Wsp pathway is a chemotaxis-like system (*Goymer et al., 2006*; *Güvener and Harwood, 2007*; *Römling et al., 2013*; *Micali and Endres, 2016*) composed of seven genes with the first six genes (*wspA-wspF*) being transcribed as a single unit and the last (*wspR* from its own promoter (*Bantinaki et al., 2007*). WspA (PFLU1219) is a methyl-accepting chemotaxis (MCP) protein that forms a complex with the CheW-like scaffolding proteins WspB (PFLU1220) and WspD (PFLU1222). WspA senses environmental stimuli and transmits the information via conformational changes in the WspA/WspB/WspD complex to effect activity of WspE (PFLU1223), a CheA/Y hybrid histidine kinase response regulator. WspE activates both the WspR (PFLU1225) diguanylate cyclase (DGC) and the CheB-like methylesterase WspF (PFLU1224) following transference of an active phosphoryl group. The activity of WspA is modulated by methylation by the constitutively active CheR-like methyltransferase WspC (PFLU1221) that transfers methyl groups to conserved glutamine residues on WspA. The demethylase WspF serves to remove these groups when in the phosphorylated active form. WS mutants are known to arise by mutations in the WspF negative regulator and also in the WspE kinase (*McDonald et al., 2009*). In vitro manipulations of WspR that abolish repression of the DGC domain by the response regulator domain are known, but have never been observed to occur in experimental populations (*Goymer et al., 2006*).

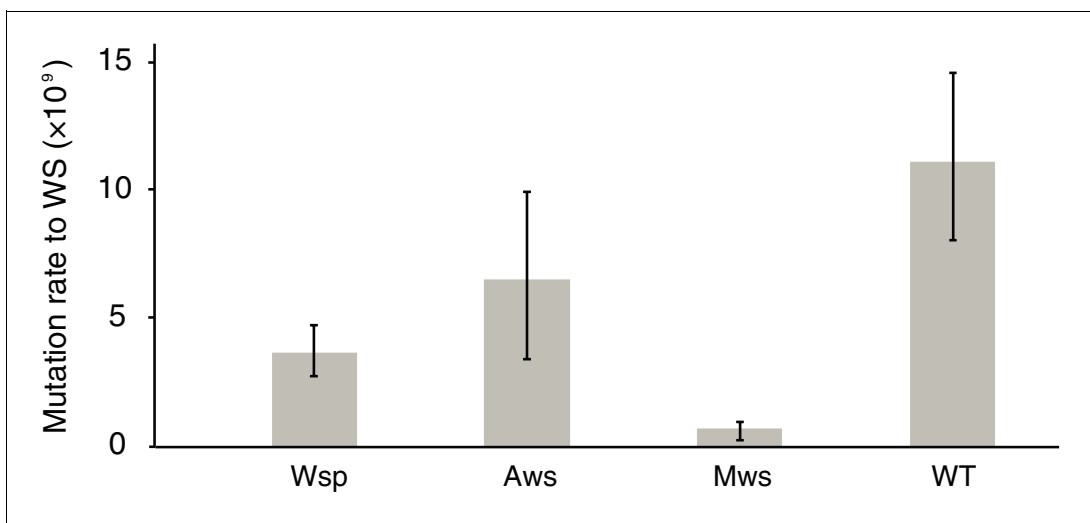

**Figure 2.** Mutation rates to WS. Fluctuation tests were used to estimate the mutation rate to WS for the three common mutational pathways to WS. Error bars represent mean and 95% confidence intervals. All mutation rates are significantly different from all others (*t*-test p<0.05, see Materials and method section for details). Number of replicates *n* = 200 for Aws and Wsp, *n* = 400 for Mws, *n* = 100 for WT.
DOI: https://doi.org/10.7554/eLife.38822.004

The 2.3 kb *aws* operon contains three genes transcribed from a single promoter (*awsXRO*). Homologous genes in *Pseudomonas aeruginosa* (*yfiRNB,* PA1121-1119) have been characterised in detail (*Malone et al., 2010*; *Malone et al., 2012*; *Xu et al., 2016*). The outer membrane lipoprotein AwsO (PFLU5209) has an OmpA domain, a signal peptide and binds to peptidoglycan. AwsO is thought to be the sensor whose activity is modulated in response to envelope stress (*Malone et al., 2012*). AwsO sequesters the periplasmic protein AwsX (PFLU5211) at the outer membrane. AwsX functions as a negative regulator of the DGC AwsR (PFLU5210) in the inner membrane. Both increased binding of AwsX to AwsO or loss of negative regulation by inactivation of the interaction between AwsX and AwsR can lead to WS (*McDonald et al., 2009*; *Malone et al., 2010*; *Malone et al., 2012*).

The 3.9 kb *mwsR* gene (PFLU5329) is known as *morA* (PA4601) in *Pseudomonas aeruginosa*, and encodes a predicted membrane protein with both a DGC domain that produces c-di-GMP and a phosphodiesterase (PDE) domain that degrades c-di-GMP. Little is known of the molecular details determining its function, but both catalytic domains appear to be active (*Phippen et al., 2014*). Deletion of the PDE domain results in a WS phenotype with activity being dependent on a functional DGC domain (*McDonald et al., 2009*).

## Development of mechanistic models predicting mutational routes to WS

If the specific effects of changing each nucleotide (and sets of nucleotides) were known then models for each pathway would not be required. Here, we show how knowledge of genetic architecture can be used to build models that predict the likelihood that mutations generating WS types will arise in a given pathway – and even in specific genes. In the following section we present four null models that incorporate increasing levels of information concerning the genotype-to-phenotype map. The goal of these models is two-fold: firstly to demonstrate that incorporation of knowledge of genetic architecture allows development of models with explanatory value and secondly, to define minimal necessary information for reliable forecasting. The results are summarised in *Figure 3*, which displays the experimental data from *Figure 2* (*Figure 3A*), along with predictions from each of the null models.

Null Model I is intentionally naïve. It uses sequence length as a proxy for mutational target size, but ignores genetic organisation, function of predicted proteins and interaction among proteins. The model assumes that mutational target size is proportional to the number of nucleotides at a given locus and thus the probability that a given pathway is used to generate WS relative to another is simply the ratio of the probability of generating WS for the two focal pathways. For any one pathway the probability that a mutation generates WS is given by $1-(1-p)^n$, where $p$ is the probability of a mutation at a nucleotide and $n$ is the number of nucleotides in the pathway. If the mutation probability is low such that the expected number of mutations in a pathway is below 1, that is, $np \ll 1$ then the binomial approximation can be used: $1-(1-p)^n = np$. Thus the probability that evolution follows the Wsp pathway over the Aws pathway is: $8400\ p_1/\ 2300\ p_2 = 3.65\ p_1/p_2$, where $p_1$ and $p_2$ are the mutation rates for each pathway. Assuming equivalency of mutation rate, $p_1 = p_2$, evolution is predicted to proceed via the Wsp pathway 3.65 times more often than via Aws, with evolution predicted to proceed via Mws 1.65 times more often than Aws. Comparison with experimental data shows a departure both in terms of the priority of pathways used by evolution and the frequency of pathway usage (*Figure 3A* versus 3B).

Null Model II builds on Model I but only in a marginal sense. It recognises that nucleotides defining loci of interest are organised into genes, and therefore adopts gene number as a proxy for mutational target size. As above, the probability that a pathway is used is $1-(1-p)^n = np$, but in this instance $n$ is the number of genes. Model II predicts that mutations in Wsp generate WS types 2.33 times more often than mutations in Aws, with mutations in Mws being 3-fold less likely to generate WS compared to mutations in Aws (*Figure 3C*). This marginal adjustment makes little difference to the fit between experimental data and predictions.

Past work has shown the explanatory value of information that comes from knowledge of gene (protein) function and interactions (*McDonald et al., 2009*; *Lind et al., 2015*). These form the basis of Null Model III. The relevant functions and interactions are depicted in *Figure 4* as reaction diagrams that reflect how changes in different interactions affect the production of WS types. Organizing interactions within each pathway according to reactions has the advantage that it allows for a

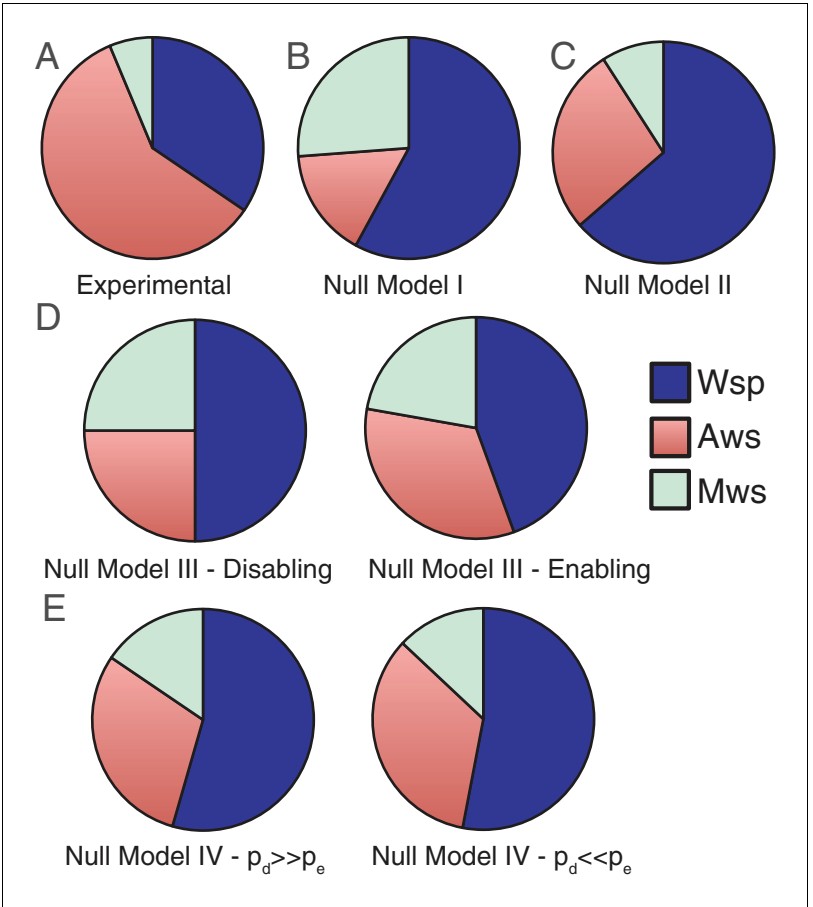

**Figure 3.** Comparisons of experimental data and null model predictions for the use of the Wsp, Aws and Mws pathways. (A) Experimental data from *Figure 2*. (B) Null Model I: number of base pairs as proxy for mutational target size. (C) Null Model II: number of genes as a proxy for mutational target size. (D) Null Model III: function and interactions between components determine mutational target size. To indicate the range of possible states we consider the situation with enabling mutations only (left hand panel) and disabling mutations only (right hand panel). (E) Null Model IV: as per null Model III, but interactions have both pleiotropic and continuous effects. To indicate the range of possible states we consider the probability of disabling mutational effects ($p_d$) to be 10 times more common than enabling changes ($p_e$), $p_d$ = 0.001, $p_e$ = 0.0001 (left hand panel) the probability of enabling mutational effects ($p_e$) to be 10 times more common than disabling changes ($p_d$), $p_d$ = 0.0001, $p_e$ = 0.001.
DOI: https://doi.org/10.7554/eLife.38822.005

standard mathematical description of the biochemical dynamics using differential equations (see *Figure 4* and *Figure 4—figure supplement 1* to 3,). An additional advantage is that such an approach allows ready incorporation of new experimental data including interaction type and interaction strength.

One immediate consequence of this approach is that production of WS types is described entirely through biochemical reactions. Knowing whether a WS-type is generated by a particular mutation amounts to determining which reaction rates are altered by the mutation and whether those changes affect the likelihood of producing WS types. This leads to an approach where mutations in components can be classified according to their effects on reaction rates: *enabling* mutations increase reaction rates, whereas *disabling* mutations decrease reaction rates.

Based on the mathematical models for the pathways, each reaction rate can be altered (either increased or decreased) to generate higher levels of the key WS effector (the active DGC). For example, in the Wsp pathway, if reactions 2 or 6 experience a disabling mutation, or alternatively, if any of the reactions 1, 3, 4, or 5 experience an enabling mutation, then the level of activated (phosphorylated) WspR is increased. For now, it is assumed that increased activity of the relevant DGC

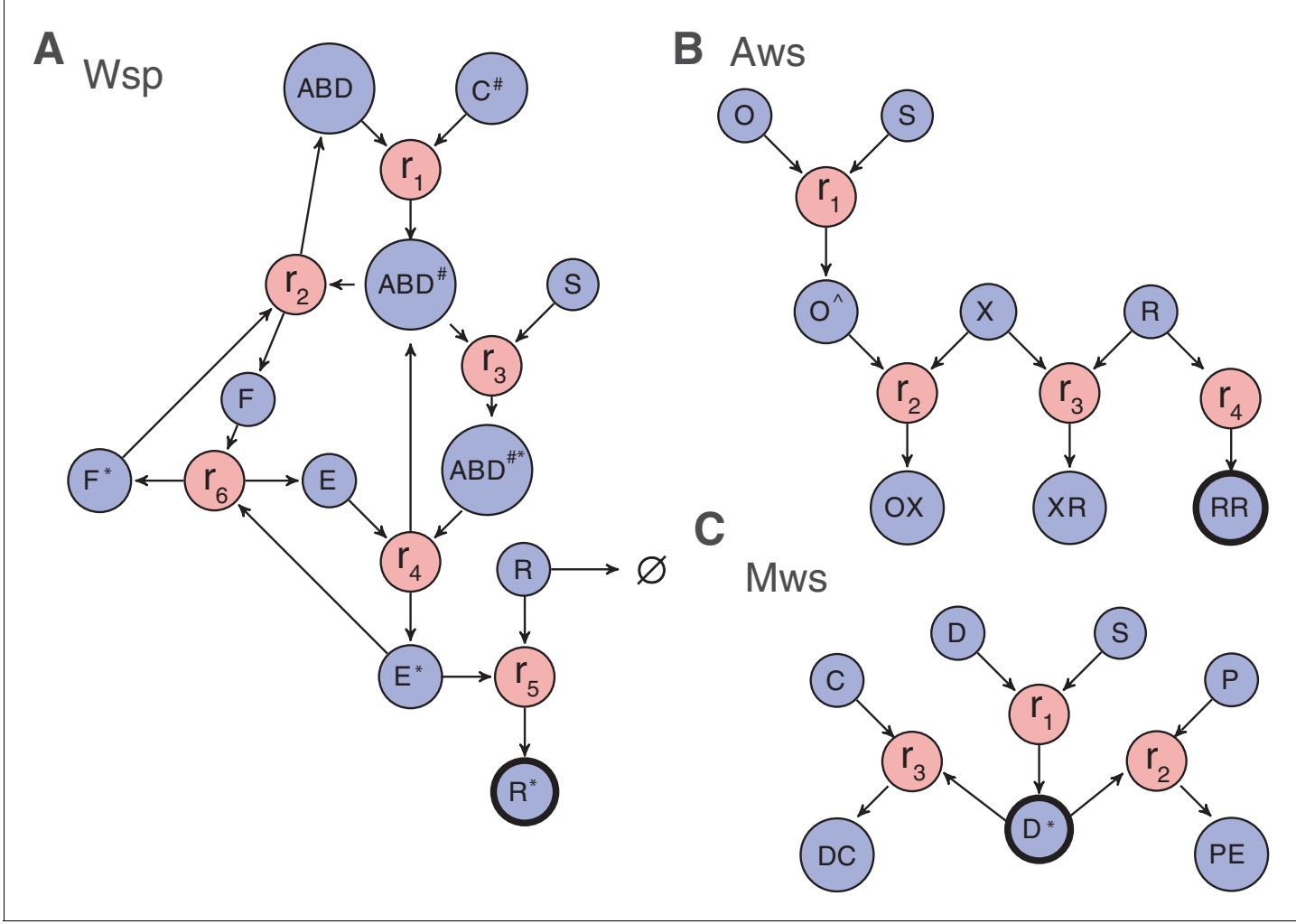

**Figure 4.** Reaction diagrams for null models III and IV. Reaction diagrams show the interactions governing production of a wrinkly spreader in (A) Wsp, (B) Aws and (C) Mws pathways. The blue circles indicate molecular components, the red circles indicate reactions, and arrows indicate which molecular components are reactants and which are products. Full descriptions of the models including all reactions and the resulting systems of differential equations are found in *Figure 4—figure supplement 1* for Wsp, *Figure 4—figure supplement 2* for Aws and *Figure 4—figure supplement 3* for Mws. The black edged circles are the reporter proteins, which is the activated forms of the diguanylate cyclases that directly determine whether a pathway generates a wrinkly spreader. Increased production of any reporter leads to increased c-di-GMP concentration causing increased cellulose production and thereby a wrinkly spreader.

DOI: https://doi.org/10.7554/eLife.38822.006

The following figure supplements are available for figure 4:

**Figure supplement 1.** Wsp model.
DOI: https://doi.org/10.7554/eLife.38822.007
**Figure supplement 2.** Aws model.
DOI: https://doi.org/10.7554/eLife.38822.008
**Figure supplement 3.** Mws model.
DOI: https://doi.org/10.7554/eLife.38822.009

generates a WS type. Reactions whose enabling/disabling mutations increase the amount of WS effector are termed enabling/disabling reactions.

By comparing the number of disabling and enabling reactions in different pathways it is possible to calculate the relative likelihood that evolution uses each mutational pathway. The Wsp pathway encompasses two disabling and four enabling reactions, whereas the Aws pathway is defined by one disabling and three enabling reactions. Mws encompasses a single disabling reaction and two

enabling reactions. If disabling mutations are as likely as enabling mutations, then the likelihood that a pathway will be used by evolution is simply the ratio of the total number of enabling and disabling reactions: 6:4:3 (Wsp:Aws:Mws). If instead, disabling mutations are much more likely than enabling mutations, then enabling reactions can be ignored and the likelihood that a pathway will be used by evolution becomes simply the ratio of the number of disabling reactions: 2:1:1 (Wsp:Aws:Mws) (*Figure 3D* left hand panel). If the reverse is true, then the likelihood of use is a ratio of the number of enabling reactions: 4:3:2 (Wsp:Aws:Mws) (*Figure 3D* right hand panel). In all cases, mutations in the Wsp pathway are predicted to be 1.3–2 times more likely to generate WS than mutations in the Aws pathway, with the Aws pathway being the target of evolution 1–1.5 times more often than the Mws pathway. Although still inflating the importance of the Wsp pathway relative to Aws, Mws is more prominent than in null Model II.

Null Model IV takes a further step toward mechanistic accuracy by endowing interactions with pleiotropic and continuous effects. This removes two simplifying assumptions of null Model III likely to limit predictive power. In Model III mutations affect one reaction at a time, but mutations in certain components, for example *wspA*, can affect more than a single reaction (WspA appears in reactions 1–4). Additionally, in Model III, changes in reaction rate are assumed to be binary, however reactions may have a range of effects on the evolution of WS types.

To accommodate pleiotropic effects, null Model IV systematically considers all combinations of enabling and disabling changes to reaction rates and determines the likelihood that a WS type is generated. An example of one set of the possible mutations ($m_i$) in Wsp is 1,–1, 0, 0, 0, 0 (an increase in $r_1$, a decrease in $r_2$, but no change in $r_3$, $r_4$, $r_5$, or $r_6$ (*Figure 3A*)). Since the Wsp pathway has six reaction rates this amounts to $3^6$ or 729 total combinations. However, note that reaction 3 does not share any reactants or products with reactions 5 or 6. Thus, mutations such as 0, 0, 1, 0, 1, 0 or 0, 0, 1, 0, 0, 1 are not considered because they require mutations in two separate genetic components.

To accommodate a range of effects null Model IV simulates enabling/disabling changes of different magnitudes and determines the resulting effect on the respective effector DGC (see Materials and methods). Briefly, the approach addresses the lack of information concerning biochemical reaction rates and molecular concentrations in the mathematical models describing WS-producing pathway dynamics. By repeatedly sampling from the space of all possible reaction rates, initial concentrations, and magnitudes of effects, this approach computes the probability that a particular set of mutations ($m_i$), for example 1,–1, 0, 0, 0, 0, results in a wrinkly spreader. This probability is represented as the conditional probability $P(WS|m_i \in Wsp)$, which motivates a Bayesian formulation to compare the relative probability that the different pathways produce WS. To this end, the probability that a particular pathway will be used is decomposed into two terms: the probability that a particular set of mutations ($m_i$) occurs in Wsp (or Aws, or Mws) represented as $P(m_i \in Wsp)$ and the probability that those mutations give rise to a wrinkly spreader represented as $P(WS|m_i \in Wsp)$ (or Aws, or Mws).

$$P(WS \cap m \cap Wsp) = \sum_i P(WS|m_i \in Wsp)P(m_i \in Wsp) \tag{1}$$

To estimate $P(m_i \in Wsp)$ we assume fixed probabilities of enabling and disabling changes and compute the product. Thus, the probability of $m_i = 1,–1, 0, 0, 0, 0$ is $p_e p_d (1 - p_e - p_d)^4$, where $p_e$ is the probability of a mutation with an enabling effect and $p_d$ is the probability of a mutation with a disabling effect. Recognising the value of accommodating the possibility of localised mutational bias we note that $p_e$ and $p_d$ can be adjusted for the affected reactants. The second term, $P(WS|m_i \in Wsp)$, relies on our sampling methodology and describes the probability that a set of disabling/enabling changes of different magnitudes will yield a WS type (see Materials and methods).

Despite the mechanistic advances incorporated into null Model IV the Wsp pathway is still predicted to be the pathway most commonly used by evolution. The extreme cases in which disabling mutations are more probable than enabling mutations (and vice versa) are shown in *Figure 3E*. The results of simulations for the full range of $p_d$ and $p_e$ values are shown in *Figure 5*. *Figure 5A* shows that the Wsp pathway is predicted to be the target of mutation 1.5–1.9 times more often than the Aws pathway while *Figure 5B* shows that the Mws pathway is predicted to be the target of mutation 0.4–0.5 times less often that the Aws pathway. While these results agree with the experimental data showing Mws to be least likely pathway to be followed, the predictions are at odds with the data

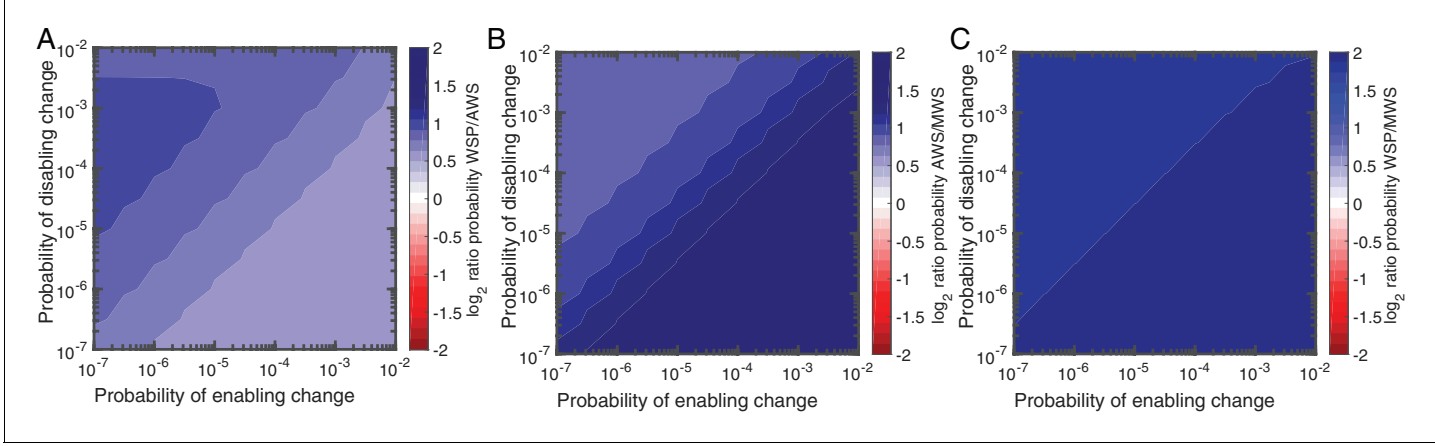

**Figure 5.** Null Model 4 predictions of the probability of using the Wsp, Aws and Mws pathways. (**A**) Probability of Wsp relative to Aws. (**B**) Probability of Mws relative to Aws. (**C**) Probability of Mws relative to Wsp. The relative contributions of individual reactions rates are available in *Figure 5—figure supplement 1* for Wsp and *Figure 5—figure supplement 2* for Aws. Sensitivity analysis is shown in *Figure 5—figure supplement 3*.

DOI: https://doi.org/10.7554/eLife.38822.010

The following figure supplements are available for figure 5:

**Figure supplement 1.** Relative contributions of reaction rates for Wsp for null Model IV.

DOI: https://doi.org/10.7554/eLife.38822.011

**Figure supplement 2.** Relative contributions of reaction rates for Aws for null Model IV.

DOI: https://doi.org/10.7554/eLife.38822.012

**Figure supplement 3.** Parameter sensitivity analysis.

DOI: https://doi.org/10.7554/eLife.38822.013

showing WS types to be twice as likely to arise from mutation in Aws, versus Wsp. The causes of this discrepancy are accounted for in the next section 'analysis of mutants reveals sources of mutational bias'.

The Bayesian approach of null Model IV has additional utility: it allows prediction of genes likely to be affected by mutation. Predictions as to specific genetic targets come from appraisal of the relative importance of each reaction in terms of generating WS types (*Figure 5—figure supplements 1* and *2*). While it is recognised that a majority of WS mutations arise from defects in negative regulators of DGCs, such as WspF and AwsX (*McDonald et al., 2009*; *Lind et al., 2015*), further predictions are possible based on impacts of alterations in gene function on reaction rates. For example, with reference to the Wsp pathway (*Figure 5—figure supplement 1*), there are two reactions (2 and 6) that are affected by WspF function: $r_2$ describes the rate of removal of methyl groups from the WspA signalling complex and $r_6$ the rate at which WspF is activated by transfer of active phosphoryl groups from the WspE kinase. Loss-of-function (disabling) mutations being much more common than gain-of-function (enabling) mutations means that WspF, WspA and WspE are all likely targets. The null model predicts that in the area of parameter space in which Wsp is most likely compared to Aws, 45% of the time WS will be generated when the second reaction, $r_2$, is altered (*Figure 5—figure supplement 1*). The same is true for reaction $r_6$. Thus the presence of a negative regulator is predicted to extend the mutational target size well beyond the gene itself. This is also true for Aws, where $r_3$ is the main contributor to the WS phenotype in the case where disabling change is more common than enabling change. Here mutations are predicted not only in the negative regulator AwsX, but also in the interacting region of the DGC AwsR (*Figure 5—figure supplement 2*).

Loss-of-function mutations in negative regulators and their interacting partners are not the only predicted targets. For Wsp $r_1$, $r_3$, $r_4$, and $r_5$ are altered approximately 5% of the time in the parameter region where disabling mutations are more common than enabling mutations, but contribute more when the rate of enabling mutations is increased (*Figure 5—figure supplement 1*). Enabling mutations based on null Model IV are likely to be found in WspC (increasing $r_1$), WspABD (increasing $r_3$), WspABD/WspE (increasing $r_4$) and WspR (increasing $r_5$) (*Figure 4A*). For Aws, enabling mutations are predicted to increase reaction $r_1$ by mutations causing constitutive activation of AwsO, $r_2$

increasing binding of AwsO and AwsX and $r_4$ increasing formation of the active AwsR dimer (**Figure 4B**, **Figure 5—figure supplement 2**).

In summary, high rates of WS mutations are predicted for *wspF, wspE, wspA, awsX* and *awsR* with lower rates for *wspC, wspR* and *awsO*. Several of these predictions sit in accord with previous experimental observations, however, notable are predictions that evolution might also target *wspA* and *wspR*, two genes that have not previously been identified as mutational causes of WS types (**McDonald et al., 2009**).

## Analysis of mutants reveals sources of mutational bias

There are several reasons why predictions from the models might be out of kilter with experimental data on mutation rates. We firstly looked to the distribution of WS generating mutations among the 109 mutants collected during the course of the fluctuation assays. Of the 109 mutants, 105 harboured a mutation in *wsp* (46 mutants), *aws* (41 mutants) or *mws* (18 mutants) (**Figure 6A**, **Figure 6—source data 1**). The remaining four had mutations in previously described rare pathways (PFLU0085, PFLU0183), again confirming that these non-focal pathways produce just a fraction of the total set of WS mutants (**Lind et al., 2015**).

The distribution of mutations for each of the three pathways is indicative of bias. As shown in **Figure 6B**, almost 29% of all WS-causing mutations (adjusted for differences in mutation rates between the three pathways) were due to an identical 33 base pair in-frame deletion in *awsX* (Δt229-g261, ΔY77-Q87), while a further 13% were due to an identical mutation (79 a->c, T27P) in *awsR*. At least 41 different mutations in Aws lead to WS: if mutation rates were equal for these sites the probability of observing 20 identical mutations would be extremely small. In fact 10 million random samplings from the observed distribution of mutations failed to recover this bias. While the Wsp pathway also contains sites that were mutated more than once (six positions were mutated twice, one site three times and one five times), sources of mutational bias in Wsp were less evident than in Aws (**Figure 6B**).

## Incorporating mutational bias into null models

The mathematical models presented above assumed no mutational bias, but the null models can be modified to incorporate such bias. With focus on the AwsX hotspot, we show this for models II-IV.

### Null Model II

The probability of a mutation at the hotspot is $kp$ where $k > 1$ and $p$ is the probability of a mutation in non-hotspot locations. The probability that a pathway is used is: $kp + (1-kp)(1-(1-p)^{(n-1)})$ where $kp$ is the probability of a mutation at the hotspot and $(1-kp)(1-(1-p)^{(n-1)})$ is the probability there is no mutation at the hotspot, but that there is a mutation elsewhere in the pathway. Using the binomial approximation, the probability a given pathway is used is: $kp + (1-kp)(n-1)p$, which for Aws is $kp+(1-kp)2p$. **Figure 7A** shows that if the mutation rate at the hotspot increases 10-fold ($k = 10.3$) then Aws is used 1.8 times more often than Wsp. While this matches experimental data, the same 10-fold increase applied to Mws results in mismatch. Only with a reduced mutation rate at the same locus ($k = 6.8$) do predicted and observed data agree (**Figure 7B**).

### Null Model III

The mutational hotspot in the Aws pathway occurs in AwsX, which is involved in both an enabling and a disabling reaction. If the hotspot increases the rate of a disabling reaction by a factor $k$ where $k > 1$, then the probability the pathway is used is:

$$k\,p_d + (1 - k\,p_d)\,(1 - (1 - p_d)_d^{(n-1)}) \; + \; (1 - k\,p_d)\,(1 - p_d)_d^{(n-1)}\,(1 - (1 - p_e)_e^n),$$

which is the sum of probabilities of the events the hotspot is used, the hotspot is not used but a disabling change occurs, and the hotspot is not used but only an enabling change occurs. **Figure 7C** shows that if $k$ is between 4 and 5.3, then there is a good fit to experimental data. The actual value depends on the probabilities of enabling/disabling change and which reaction has the hotspot. For Mws the mutational hotspot must have a factor between 10 and 15 (**Figure 7D**).

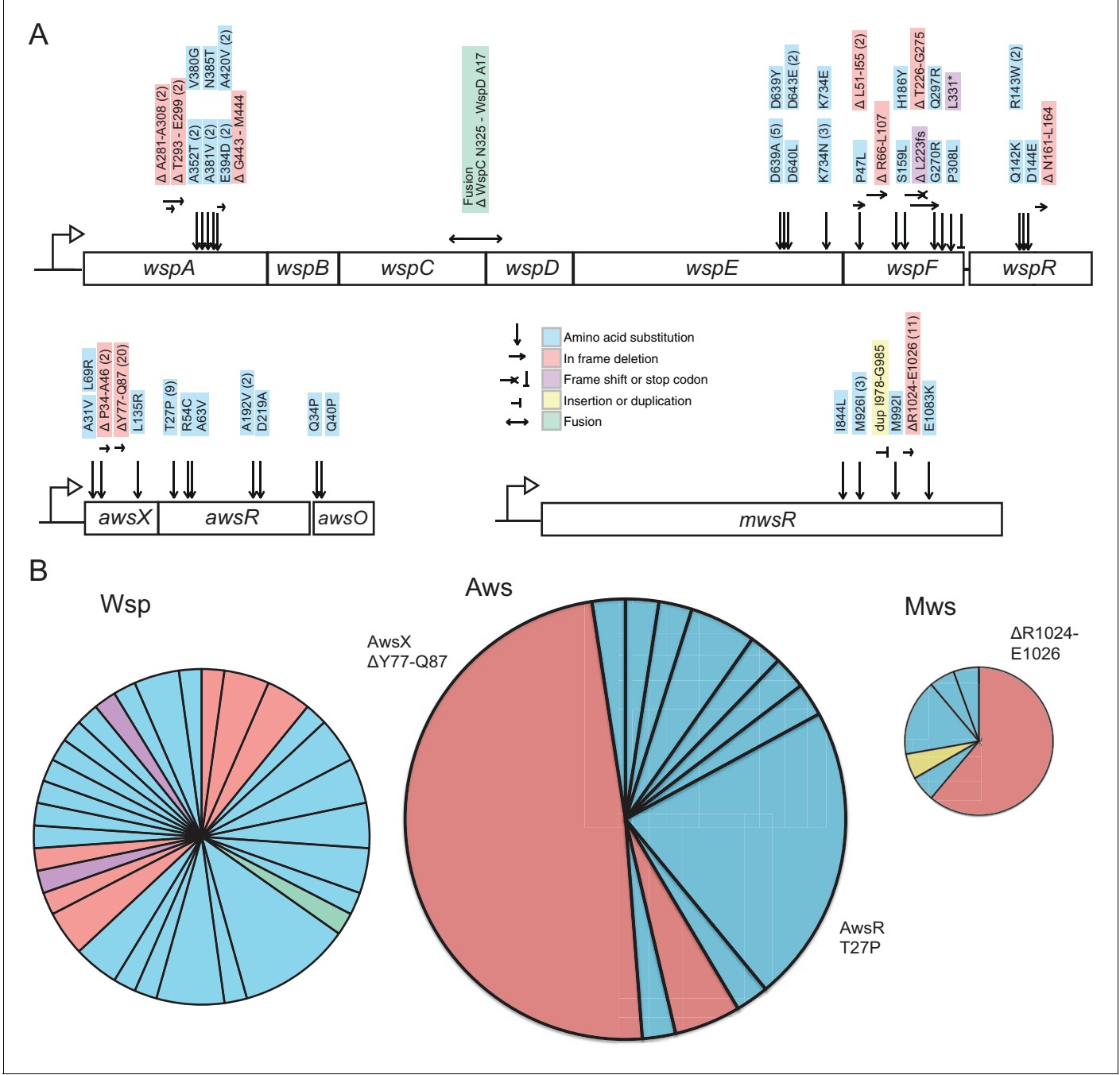

**Figure 6.** Mutational targets. (**A**) 105 independent mutations in the *wsp* (n = 46), *aws* (n = 41) and *mws* (n = 18) operons were identified. Numbers of independent mutants are shown in brackets. Full details on the mutations are available in Figure 5—source data 1. (**B**) Diversity of mutations with area proportional to mutation rate (***Figure 2***). Two mutations (AwsX ΔY77-Q87 and AwsR T27P) contribute 41% of all mutations to WS suggesting that these are mutational hot spots.

DOI: https://doi.org/10.7554/eLife.38822.014

The following source data is available for figure 6:

**Source data 1.** Table of all WS mutations in Wsp, Aws and Mws.
DOI: https://doi.org/10.7554/eLife.38822.015

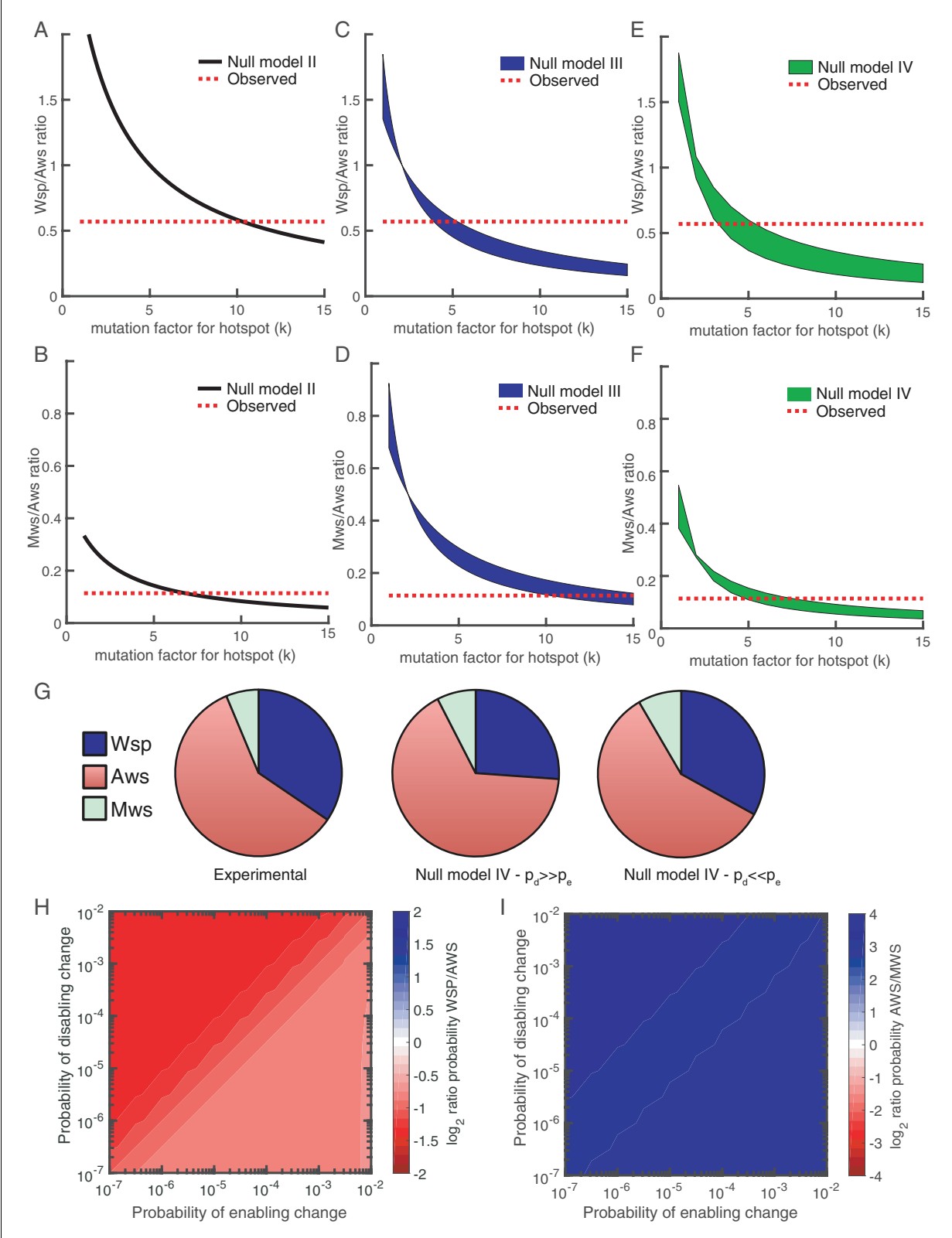

**Figure 7.** Revised models including mutational hot spot. Null Models II (**A, B**), III (**C, D**) and IV (**E, F**) can be revised to take into account the mutational hot spot in AwsX by including a mutation factor ($k$). Each plot (**A-F**) shows the predicted ratios of Wsp/Aws or Aws/Mws compared to the experimental results (dashed red line). For null models III and IV a range is shown that depends on the probabilities of enabling ($p_e$) and disabling ($p_d$) changes. (**G**) Comparison of experimental data (left) and revised predictions from null model IV (middle, right), incorporating a five-fold increase in $p_e$ and $p_d$ for $r_2$

*Figure 7 continued on next page*

*Figure 7 continued*

and $r_3$ in the Aws differential equation system. Middle pie chart use $p_d$ = 0.001, $p_e$ = 0.0001 and right pie chart $p_d$ = 0.0001, $p_e$ = 0.001 to allow comparison to *Figure 3F and G*. (H) Null Model IV predicted probability of Wsp relative to Aws with a hotspot in AwsX that increases the mutation rate five-fold for $p_e$ and $p_d$ for $r_2$ and $r_3$ in the Aws system of differential equation (I) Null Model IV predicted probability of Aws relative to Mws with a hotspot in AwsX that increase the mutation rate five-fold for $p_e$ and $p_d$ for $r_2$ and $r_3$ in the Aws system of differential equation. *Figure 7—figure supplement 1* shows the relative contribution of each reaction rate in the Aws network to the production of WS for the revised null model IV with a 5 times increase for $p_e$ and $p_d$ for $r_2$ and $r_3$.

DOI: https://doi.org/10.7554/eLife.38822.016

The following figure supplement is available for figure 7:

**Figure supplement 1.** Relative contributions of reaction rates for Aws for null Model IV with a five-fold increase in mutation factor for $r_2$ and $r_3$.

DOI: https://doi.org/10.7554/eLife.38822.017

## Null Model IV

The mutational hotspot in *awsX* affects reaction rates $r_2$ and $r_3$ in the Aws system of differential equations (*Figure 4B*, *Figure 4—figure supplement 2*). If the mutation factor is between 3.3–5.6 for the Wsp/Aws comparison (*Figure 7E*) or 4.8–7.6 for the Mws/Aws comparison (*Figure 7F*), then predictions match observed data. Thus Model IV is the only one where the same five-fold change in the probability of enabling/disabling change predicts observed data for all three mutational pathways (*Figure 7G*). This model, updated with hotspot data, predicts that the Aws pathway is more likely to generate WS types than Wsp for all probability values (see *Figure 7H*). If disabling changes are more probable than enabling changes, then $r_3$ dominates the predicted rate, whereas when enabling changes are more common $r_2$ is the major contributor (*Figure 7—figure supplement 1*). Knowledge of the functional interaction between AwsX and AwsR allows interpretation of the likely effect of the hotspot mutations in AwsX and AwsR as disabling change in $r_3$ (*Table 1*).

## Analysis of mutants reveals mutational targets and effects

Null Model IV of the Wsp, Aws, and Mws pathways allow estimation of the relative probability that a WS is produced by enabling or disabling changes in a certain reaction rate. This means that in order to evaluate the success of our predictions, the 105 WS mutations must be connected to their likely effects on reaction rates. For some mutations this is straightforward. If a mutation completely disables the negative regulator WspF, this will reduce $r_2$ and $r_6$, thereby producing a wrinkly spreader. In other cases interpretations of likely effects are more difficult and requires knowledge of the molecular functions of the proteins and their interactions, which needs to be obtained from literature, bioinformatics and structural modelling. For example, mutations in WspA can potentially produce WS by disruption of the interaction with WspF (reducing $r_2$), increase methylation by WspC (increasing $r_1$), increase signalling rate ($r_3$) or increase phosphorylation of WspE ($r_4$). A summary of the interpretation of the molecular effects of mutations and how they are connected to reaction rates is available in *Table 1* with a more detailed description in Appendix 1.

Mutations were found in five genes in the Wsp pathway. The majority of the mutations were found in the negative regulator WspF or its interacting partners WspE and WspA. These mutations likely reduced reaction rates of $r_2$ and $r_6$ in accord with predictions of null Model IV. The remaining mutations in WspC and WspR were less common, in accord with predictions and likely due to increases in reaction rates $r_1$ and $r_5$, respectively. Results for the Aws pathway were also in line with predictions with the majority of the mutations arising in the negative regulator AwsX or the interacting part of AwsR (both decreasing $r_3$), with rare mutations in AwsO and other parts of AwsR. For MwsR, loss-of-function mutations were expected primarily in the phosphodiesterase (PDE) domain, but mutations were found in both diguanylate cyclase (DGC) and PDE domains. A structural analysis revealed that mutations were clustered in the interface between the domains and unlikely to disrupt PDE function. These mutations most likely change the dynamics between the DGC and EAL domains in a way that increases production of c-di-GMP.

## Differences of mutational spectra with and without selection

The null models – and especially null Model IV – successfully predicted the mutational targets and explained mutation rates to WS when knowledge of mutational hot spots was included. To what

**Table 1.** Connection of mutational effects to reaction rates

| Protein | Mutation class | Proposed molecular effect | Proposed reaction rate effect | References |
|---|---|---|---|---|
| WspA | Amino acid substitutions 352–420 | Trimer-of-dimer formation, localisation of Wsp clusters, interaction with WspD | Increase $r_3$ | (*O'Connor et al., 2012*; *Griswold et al., 2002*) |
| WspA | Deletions A281-A308 | Disrupt demethylation by WspF | Decrease $r_2$ | (*McDonald et al., 2009*) |
| WspC/D | Fusion of WspC and WspD | Increased methylation by WspC, blocking access of WspF | Increase $r_1$, decrease $r_2$ | |
| WspE | Response regulatory domain phosphorylation site | Disrupt phosphorylation of WspF | Decrease $r_6$ | Homology model |
| WspF | Any disabling mutation | Loss of negative regulation by WspF by demethylation of WspA | Decrease $r_2$ | (*Bantinaki et al., 2007*; *McDonald et al., 2009*) |
| WspR | Amino acid substitutions and small deletion in linker region between response regulator and DGC domain | Constitutive activation of DGC without phosphorylation. Effects on subcellular clustering or multimeric state. | Increase $r_5$ | (*Goymer et al., 2006*; *De et al., 2009*; *Huangyutitham et al., 2013*) |
| AwsX | In frame loss of function mutations | Loss of negative regulation of AwsR | Decrease $r_3$ | (*Malone et al., 2012*) |
| AwsR | Amino acid substitutions transmembrane helix, periplasmic domain | Disruption of interaction with AwsX | Decrease $r_3$ | (*Malone et al., 2012*) |
| AwsR | Amino acid substitutions HAMP linker domain | Changes in dimerization, packing of HAMP domains causing constitutive activation | Increase $r_4$ | (*Parkinson, 2010*; *Malone et al., 2012*) |
| AwsO | Amino acid substitutions in between signal peptide and OmpA domain | Constitutive activation and sequestering of AwsX | Increase $r_2$ | (*McDonald et al., 2009*; *Malone et al., 2012*; *Xu et al., 2016*) |
| MwsR | Amino acid substitutions and small in frame deletions in interface between EAL and DGC domains | Changes to interdomain interaction or dimerization leading to constitutive activation of DGC function | Decrease $r_2$, increase $r_1$ | Homology model, (*McDonald et al., 2009*) |
| MwsR | Amino acid substitutions near DGC active site | Loss of feedback regulation or changes to interdomain interaction | Decrease $r_2$, increase $r_1$ | Homology model, (*McDonald et al., 2009*) |

DOI: https://doi.org/10.7554/eLife.38822.018

degree can such knowledge allow prediction of the outcome of the original experimental evolution under selection (*McDonald et al., 2009*)? A comparison between the frequencies of mutations isolated here (without selection) with those isolated under selection, reveals several notable differences (*Figure 8*). The most obvious difference is in use of the Wsp pathway, which is most commonly used (15/24) under selection and yet produces WS types at a lower rate than the Aws pathway. Differences are also apparent in the spectrum of *wsp* mutations, with no *wspA* mutations being found under selection despite being the most commonly mutated gene without selection (15/46), and the previous failure to detect *wspR* mutants in a screen of 53 WS mutants (*Goymer et al., 2006*).

## Fitness of WS types

The most obvious explanation for the differences in mutational spectra between WS isolated with or without selection (*Figure 8*) is that certain mutants have a lower fitness and thus their relative

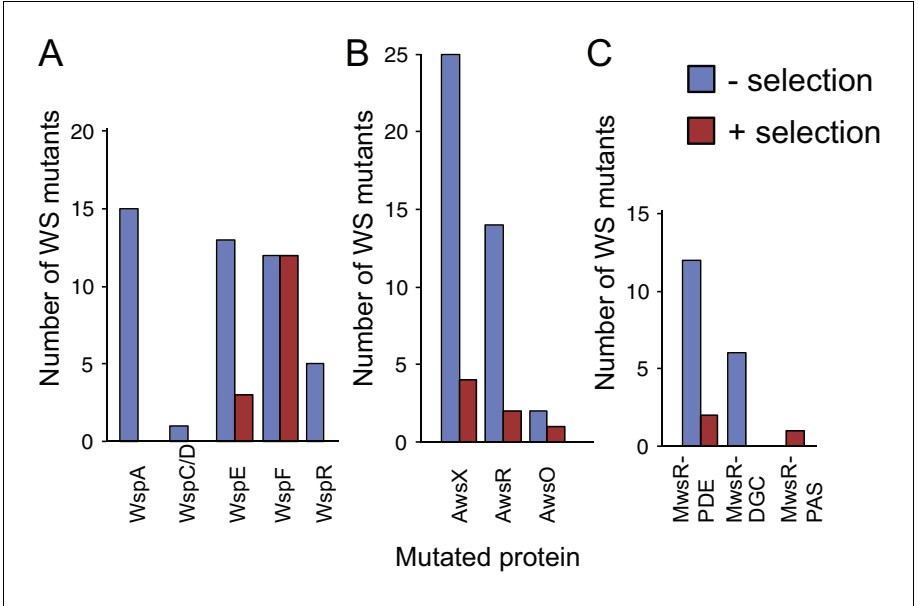

**Figure 8.** WS mutations isolated with and without selection. (**A**) The mutational spectrum observed under selection suggests underrepresentation of WspA and WspR compared to WspF and WspE. (**B**) Similar fitness effects of different Aws mutants lead to similar patterns regardless of selective conditions. However the frequency of all Aws mutants isolated under selection is much lower than expected from its high mutation rate. (**C**) The low number of MwsR mutants isolated does not allow an analysis of relative rates. Only within operon comparisons are valid for this figure as the mutants isolated without selection had double deletions of the other operons. Between operon mutation rates are available in *Figure 2*.
DOI: https://doi.org/10.7554/eLife.38822.019

frequencies will be lower in the original evolution experiment (*McDonald et al., 2009*). We measured the fitness of representative WS types with mutations in each of the mutated genes (*wspA*, *wspC/D*, *wspE*, *wspF*, *wspR*, *awsX*, *awsR*, *awsO*, *mwsR*) in 1:1 competitions against a reference WspF ΔT226-G275 deletion mutant marked with GFP (*Figure 9*). This type of fitness data should be interpreted with caution because the fitness of WS mutants are frequency-dependent (*Rainey and Travisano, 1998*) and some WS mutants are superior in early phase attachment as opposed to growth at the air-liquid interface (*Lind et al., 2015*). Nevertheless, the competition experiments provide an estimate of fitness when several different WS mutants compete at the air-liquid interface (a likely situation given a ~ $10^{-8}$ mutation rate to WS and a final population size of >$10^{10}$). The fitness data account for the over- or under-representation of some WS mutants when grown under selection (*McDonald et al., 2009*) compared to those uncovered without selection (as reported here).

The three *wspF* mutants, the *wspC-wspD* fusion, and the *wspE* mutants have similar fitness (p>0.38, except for WspE K734E that has slightly higher fitness (p=0.027), two-tailed *t*-tests). In contrast, both *wspA* mutants are slightly less fit (p<0.0214, two-tailed *t*-test) and both *wspR* mutants are severely impaired (p<0.00007, two-tailed *t*-test) (*Figure 9*). This sits in accord with previous work in which mutations generating WS obtained with selection have been detected in *wspF* and *wspE*, but not *wspA* or *wspR* (*Goymer et al., 2006*; *McDonald et al., 2009*). All *awsXRO* mutants have similar lower fitness (p<$10^{-6}$, two-tailed *t*-test) compared to the *wspF* reference strain (*Figure 9*), which explain why under selection these are found at lower frequencies compared to mutations in the *wsp* pathway (*McDonald et al., 2009*) despite a roughly two-fold higher mutation rate to WS.

## Discussion

The issue of evolutionary predictability and the relative importance of stochastic events compared to deterministic processes have a long history in evolutionary biology (*Darwin, 1872*; *Simpson, 1949*; *Jacob, 1977*; *Gould, 1989*; *Conway Morris, 2003*; *Orgogozo, 2015*). Recent interest has been

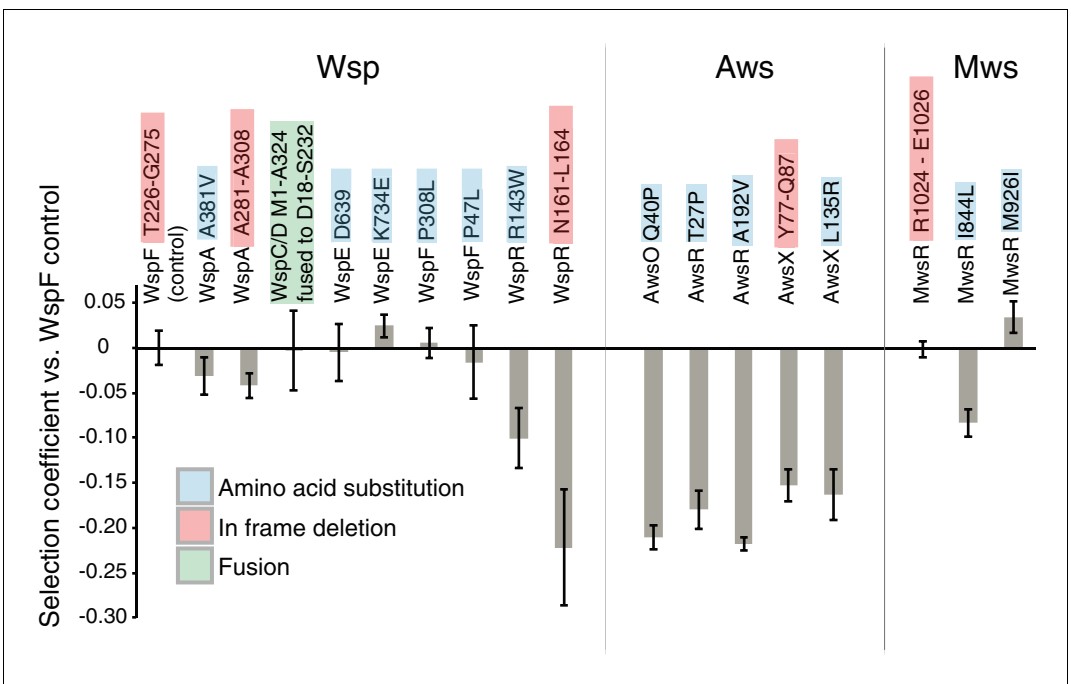

**Figure 9.** Fitness of different WS mutants. Competitive fitness against a WspF ΔT226-G275 reference strain was measured for representative mutations in the Wsp, Aws, Mws pathways. Pairwise competitions were performed in quadruplicates and error bars represent ±one standard deviation. Full data, including statistical tests are available in *Figure 9—source data 1*.

DOI: https://doi.org/10.7554/eLife.38822.020

The following source data is available for figure 9:

**Source data 1.** Data from fitness assay.
DOI: https://doi.org/10.7554/eLife.38822.021

sparked by an increasing number of observations that both phenotypic and molecular evolution, under certain circumstances, can be remarkably repeatable (*Colosimo et al., 2005*; *Shindo et al., 2005*; *Jost et al., 2008*; *Barrick et al., 2009*; *Lee and Marx, 2012*; *Meyer et al., 2012*; *Zhen et al., 2012*; *Herron and Doebeli, 2013*), but whether these cases are representative for evolutionary processes in general remains to be determined. A related question, with greater potential for practical applications, is whether it is possible to forecast short-term evolutionary events and if so, then the challenge is to stipulate the data necessary to make successful predictions.

Our uniquely detailed knowledge of the WS experimental evolution system has provided a rare opportunity to determine the contributions of mutational bias and genetic architecture to the generation of new adaptive phenotypes, and consequently explore the limits of evolutionary forecasting. A thorough understanding of the function of the molecular species and their interactions allowed development of null models (see especially Model IV) that capture essential features of the genotype-to-phenotype map sufficient to predict the relative likelihood that evolution will follow each of the three principle pathways, along with specific mutational (genetic) targets. Even the most sophisticated of these models failed to forecast outcomes that matched the experimental data shown in *Figure 2*, however, the reason became apparent upon characterisation of the set of WS mutants obtained without selection: the presence of mutational hotspots. Armed with knowledge of sources of mutational bias it was a simple matter to refine Model IV leading to predictions matching those observed by experiment.

Problematic at the current time is inability to *a priori* detect all causes of mutational bias, however, it is likely that this will improve as understanding of the biochemical causes of bias improves and algorithms trained to recognise and detect nucleotide patterns indicative of bias are implemented. One specific deletion (ΔY77-Q87) in *awsX* was found to account for nearly half (20/41) of the mutations in the Aws pathway. Thus, despite the existence of hundreds of possible mutations

leading to WS (this work and (*McDonald et al., 2009*; *McDonald et al., 2011*; *Lind et al., 2015*)) one single mutation accounts for more than one quarter of all WS mutations. While the six base pair direct repeat flanking the deletion provides a convincing explanation for its increased rate, it is not clear why this deletion would be ten times more common than the ΔP34-A46 deletion in the same gene that is flanked by ten base pair repeats and contains five base pairs identical to those from the ΔY77-Q87 deletion (*Figure 6—source data 1*). There are also instances where single base pair substitutions are overrepresented: the AwsR T27P mutation is found in nine cases, while eight other single pair substitutions in Aws were found only once. Consider further the fact that *wspE* (a gene of ~2.3 kb), where changes to only four specific amino acids repeatedly cause WS, and *wspF* (a gene of ~1 kb) where any mutation that disrupts function results in WS (*Figure 6A*) contribute equally to the rates of at which new WS types arise.

It is evident from these findings and from related studies (*Pollock and Larkin, 2004*) that there is need for detailed experimental measurement of local mutation rates in specific systems. Such investigations stand to contribute to understanding of the causes of mutational bias and the extent to which biases might be conserved among related or even unrelated organisms. If local nucleotide sequence is the major determinant, an estimate of mutation rate will apply strictly to very closely related species, but if the dynamics of molecular processes, such as transcription and replication (*Sankar et al., 2016*), are major influences then estimates might be applicable to a wider range of species.

Forecasting mutational routes to new phenotypes is one component of a comprehensive forecasting strategy. The second requirement is ability to *a priori* predict fitness effects. Solving this problem is the Holy Grail for predicting evolution, but at the current time this is not possible. That it matters is made clear by the observation that a subset of all possible WS mutants was found after experimental evolution (*Figure 8*) due to inability of some to successfully compete with high fitness WS types (*Figure 9*).

Direct measurement of the fitness effects of large numbers of mutations is difficult, time-consuming, typically only possible for microbial species and against the spirit of identifying *a priori* predictors of fitness effects. Future success likely rests on ability to infer fitness from parameters such as estimated effects of mutations on thermodynamic stability (*Capriotti et al., 2005*; *Dean and Thornton, 2007*; *Rodrigues et al., 2016*), molecular networks, evolutionary conservation of amino acid residues (*Glaser et al., 2003*; *Ng and Henikoff, 2003*; *Landau et al., 2005*; *Kumar et al., 2009*; *Celniker et al., 2013*; *Choi and Chan, 2015*) or machine-learning methods combining several types of information (*Bromberg and Rost, 2007*; *Li et al., 2009*; *Capriotti et al., 2013*; *Yates et al., 2014*; *Hecht et al., 2015*). Possibilities likely also exist to extrapolate findings from a small number of mutations that are either directly constructed and assayed in the laboratory or through fitness estimates of polymorphism data from natural populations. Recent work on the prediction of the fitness effects of random mutations in several genes suggests that in many cases large effect mutations can be predicted using methods based on evolutionary conservation (*Lind et al., 2017a*).

On a less ambitious scale it may be possible to take advantage of the fact that the distribution of fitness effects associated with mutations in single genes is often bimodal – a consequence of many mutations causing complete loss-of-function rather than intermediate deleterious; effects (*Sanjuan, 2010*; *Hietpas et al., 2011*; *Jacquier et al., 2013*; *Firnberg et al., 2014*; *Lind et al., 2017a*; *Sarkisyan et al., 2016*; *Lundin et al., 2017*). As such it may be sufficient to know gene function plus consequences arising from loss of function mutations and thus treat mutations within single genes as having equivalent fitness (*Sanjuan, 2010*; *Jacquier et al., 2013*; *Sarkisyan et al., 2016*; *Lind et al., 2017a*). Interestingly WS mutations occurring in the same gene typically show similar fitness effects (*Figure 9*). While still requiring experimental data, this simplification may fuel exploration of the relationship between large numbers of mutations and their fitness effects and thus understanding of the extent to which fitness effects are transferable between strains with different genetic backgrounds or closely related species (*Ungerer et al., 2003*; *Pearson et al., 2012*; *Wang et al., 2014*).

Taken at face value, our findings give reason to question the value of aspiring to forecast evolution from first principles based on mechanistic understanding. But we argue against such pessimism and point firstly to the value that stems from a clear understanding of current limitations on forecasting: defining what is known, what is not known, and what needs to be known. That it is possible to take knowledge of the genotype-to-phenotype map and forecast with good accuracy targets and

rates is an important advance. There is good reason to suspect that the principles outlined here and previously (*Lind et al., 2015*) are transferable to other systems and even generalizable (*Lind, 2018*).

Evolutionary forecasting is in its infancy. In the short term it is likely to be most successful for biological systems where there are experimental data on a large number of independent evolutionary events, such as is the case for influenza, HIV and cancer (*Kouyos et al., 2012*; *Fraser et al., 2014*; *Lawrence et al., 2014*; *Luksza and Lässig, 2014*; *Neher et al., 2014*; *Eirew et al., 2015*). Evolution might appear idiosyncratic indicating that every specific system requires detailed investigation, but our hope is that deeper knowledge of the genotype-to-phenotype map, distribution of fitness effects and mutational biases will allow short term forecasts to be produced using modelling without the need for large-scale experimental studies. A major boost to further refinement of evolutionary forecasting is likely to come from combining coarse (top down) and fine-grained (bottom-up) approaches. Our demonstration that simple null models of functional networks can produce quantitative predictions is a step forward allowing predictions to be directly tested in other experimental systems (*Lind, 2018*).

# Materials and methods

## Key resources table

| Reagent type (species) or resource | Designation | Source or reference | Identifiers | Additional information |
|---|---|---|---|---|
| Gene | *wspA*; PFLU1219 | NA | | |
| Gene | *wspB*; PFLU1220 | NA | | |
| Gene | *wspC*; PFLU1221 | NA | | |
| Gene | *wspD*; PFLU1222 | NA | | |
| Gene | *wspE*; PFLU1223 | NA | | |
| Gene | *wspF*; PFLU1224 | NA | | |
| Gene | *wspR*; PFLU1225 | NA | | |
| Gene | *awsX*; PFLU5211, *yfiR* | NA | | |
| Gene | *awsR*; PFLU5210, *yfiN*, *tpbB* | NA | | |
| Gene | *awsO*; PFLU5209, *yfiB* | NA | | |
| Gene | *mwsR*; PFLU5329, *morA* | NA | | |
| Gene | wss; PFLU0300-PFL0309 | NA | | |
| Strain, strain background | *Pseudomonas fluorescens* SBW25 | PMID: 19432983; PMID: 8830243 | | |
| Genetic reagent | PBR721; Δaws Δmws | PMID: 19704015 | | |
| Genetic reagent | PBR713; Δwsp Δmws | PMID: 19704015 | | |
| Genetic reagent | PBR712; Δwsp Δaws | PMID: 19704015 | | |
| Genetic reagent | WT with pMSC | PMID: 17377582 | | |
| Genetic reagent | PBR721 with pMSC | This work | | Wsp pathway with pMSC |
| Genetic reagent | PBR713 with pMSC | This work | | Aws pathway with pMSC |
| Genetic reagent | PBR712 with pMSC | This work | | Mws pathway with pMSC |
| Cell line | | | | |
| Transfected construct | | | | |
| Biological sample | | | | |
| Antibody | | | | |
| Recombinant DNA reagent | pMSC | PMID: 17377582 | | |
| Recombinant DNA reagent | mini-Tn7(Gm)P$_{rrnB\ P1}$ gfp-a | PMID: 15186351 | | |

*Continued on next page*

*Continued*

| Reagent type (species) or resource | Designation | Source or reference | Identifiers | Additional information |
|---|---|---|---|---|
| Sequence-based reagent | PFLU1219_1294R | Sigma-Aldrich | | agacactgcggactggatct |
| Sequence-based reagent | PFLU1219_UF | Sigma-Aldrich | | ttgcctgtggagcgttctat |
| Sequence-based reagent | PFLU1219_1130F | Sigma-Aldrich | | accaggtggtggtgaccat |
| Sequence-based reagent | PFLU1220_DR | Sigma-Aldrich | | gacgatcactgcttcgatca |
| Sequence-based reagent | wspD1r | Sigma-Aldrich | | gttgcgttccagggcgag |
| Sequence-based reagent | PFLU1221_DR | Sigma-Aldrich | | acttatcgccatggataccg |
| Sequence-based reagent | PFLU1221_UF | Sigma-Aldrich | | tggatcggtgtaaacgacct |
| Sequence-based reagent | PFLU1223_1511F | Sigma-Aldrich | | atgacatcgtgcaactggaa |
| Sequence-based reagent | PFLU1223_DR | Sigma-Aldrich | | caggtccatcaggatcaggt |
| Sequence-based reagent | PFLU1223_841F | Sigma-Aldrich | | cgcccaagtgttgtatgaca |
| Sequence-based reagent | PFLU1223_DR | Sigma-Aldrich | | caggtccatcaggatcaggt |
| Sequence-based reagent | PFLU1224_160R | Sigma-Aldrich | | tccatcaccggcatgatca |
| Sequence-based reagent | PFLU1224_DR | Sigma-Aldrich | | ttttcgtcggtcttgatgtc |
| Sequence-based reagent | PFLU1224_UF | Sigma-Aldrich | | aagccagtttccacgatgac |
| Sequence-based reagent | PFLU1225_DR | Sigma-Aldrich | | gtgagggtacaccgagccta |
| Sequence-based reagent | PFLU1225_UF | Sigma-Aldrich | | tgccaaatgaacatgacctc |
| Sequence-based reagent | PFLU5329_1888F | Sigma-Aldrich | | cgtggattgggtgttcaaga |
| Sequence-based reagent | PFLU5329_3118R | Sigma-Aldrich | | cggttggtaatagagcacga |
| Sequence-based reagent | wspF-seqR | Sigma-Aldrich | | ataacggtcggcttgatacg |
| Sequence-based reagent | Aws11f | Sigma-Aldrich | | gctggttcagcttgatcgaaccc |
| Sequence-based reagent | Aws16r | Sigma-Aldrich | | atggatgctgccgatggttc |
| Sequence-based reagent | Aws7f | Sigma-Aldrich | | gcaaactcaccgcggatgag |
| Sequence-based reagent | Aor | Sigma-Aldrich | | cccgtcgattctccgtgcgc |
| Peptide, recombinant protein | | | | |
| Commercial assay or kit | | | | |
| Chemical compound, drug | | | | |

*Continued on next page*

*Continued*

| Reagent type (species) or resource | Designation | Source or reference | Identifiers | Additional information |
|---|---|---|---|---|
| Software, algorithm | Ma-Sandri-Sarkar MLE estimator | PMID: 19369502 | | www.keshavsingh.org/ protocols/FALCOR.html |
| Software, algorithm | Julia v1.0 | https://julialang.org | | |
| Software, algorithm | Matlab 9.4 | | | |
| Software, algorithm | Phyre2 | PMID: 25950237 | | http://www.sbg.bio. ic.ac.uk/phyre2 |
| Software, algorithm | Geneious 7.1 | https://www. geneious.com | | |
| Software, algorithm | FACSDiva 7.0 | BD Biosciences | | |
| Other | | | | |

## Strains and media

The strains used in the study are all *Pseudomonas fluorescens* SBW25 (*Silby et al., 2009*) or derivatives thereof. The reporter construct (pMSC), used for isolation of WS mutants before selection, fused the $P_{wss}$ promoter to a kanamycin resistance marker (*nptII*) (*Fukami et al., 2007*; *McDonald et al., 2011*). *P. fluorescens* strains with deletions of the *wsp* (PFLU1219-1225), *aws* (PFLU5209-5211) and *mws* (PFLU5329) operons were previously constructed as described by McDonald *et al.* (*McDonald et al., 2011*). All experiments used King's medium B (KB) (*King et al., 1954*), solidified with 1.5% agar and incubation was at 28°C. All strains were stored in glycerol saline KB at −80°C.

## Fluctuation tests and isolation of WS mutants before selection

Strains with the pMSC reporter construct and either wild type genetic background or double deletions of *aws/mws*, *wsp/mws* or *wsp/aws* were used to estimate mutation rates to WS before selection. Overnight cultures were diluted to approximately $10^3$ cfu/ml and 60 independent 110 µl cultures were grown for 16–19 hr (OD600 = 0.9–1.0) with shaking (200 rpm) in 96-well plates before plating on KB plates with 30 mg/l kanamycin. Viable counts were estimated by serial dilution and plating on KB agar. One randomly chosen colony per independent culture with WS colony morphology was restreaked once on KB agar. The assay was repeated at least four times for the double deletion mutants and twice for the wild type strain in order to obtain enough mutants to allow estimation of mutation rates. Mutations rates and confidence intervals were estimated using the Ma-Sandri-Sarkar Maximum Likelihood Estimator (*Hall et al., 2009*) available at www.keshavsingh.org/ protocols/FALCOR.html. The mutation rates between the different strains were statistically evaluated using a t-test as previously described (*Rosche and Foster, 2000*), but this method has only been shown to be valid in cases where total population size is not significantly different for the strains used. In our assay this was not the case, as determined by ANOVA, and therefore the results of the statistical analysis should be interpreted with caution. As the estimated number of mutants per well was <0.5 for all strains, the biasing effect of differences in fitness between WS mutants is minimal.

## Sequencing

Mutations causing the WS phenotype were identified by Sanger sequencing of candidate genes in the remaining common pathway to WS, for example the *wsp* operon for the *aws/mws* deletion strain. In a few cases where no mutations were identified in the previously established WS target genes, we used genome sequencing (Illumina HiSeq, performed by Macrogen Korea).

## Fitness assays

Competition assays were performed as previously described (*Lind et al., 2015*) by mixing the WS mutant 1:1 with a reference strain labelled with green fluorescent protein and measuring the ratio of

each strain before and after static growth for 24 hr using flow cytometry (BD FACS Canto). We used a WspF ΔT226-G275 deletion mutant as the reference strain because WspF mutants are the most commonly found WS type when grown under selective conditions (*McDonald et al., 2009*) and the in frame deletion of 50 amino acids most likely represents a complete loss-of-function mutation with minimal polar effects on the downstream *wspR*. Selection coefficients per generation were calculated as s = [ln(R(t)/R(0))]/[t], as previously described (*Dykhuizen, 1990*) where R is the ratio of alternative WS mutant to WspF ΔT226-G275 GFP and t the number of generations. Viable counts on KB plates of initial and final populations were performed to calculate the number of generations. Stability of colony morphologies was confirmed and data from microcosms with >5% smooth colonies were excluded (two cases). Control competition experiments with isogenic WspF ΔT226-G275 reference strains with and without GFP were used to correct for the cost of the GFP marker. Control competitions were also used to determine the cost of the double deletions and the reporter construct relative to a wild type genetic background, for example an AwsX ΔY77-Q87 mutant in Δwsp/Δmws background with pMSC was competed with a GFP labeled AwsX ΔY77-Q87 mutant in wild type background. Competitions were performed in independently inoculated quadruplicates for each strain with the number of replicates based on previous work (*Lind et al., 2015*).

## Homology models

Homology models of the structure of WspA, WspE, WspR, AwsR, AwsO and MwsR were made using Phyre2 in intensive mode (http://www.sbg.bio.ic.ac.uk/phyre2) (*Kelley et al., 2015*).

## Probability estimation in the null model IV

The differential equation models describe the interactions between proteins in each of the three WS pathways. In order to solve the differential equations, two pieces of information are required: (i) the initial concentrations of the molecular species and (ii) the reaction rates. Although this information is unavailable a random-sampling approach was used to generate different random sets of initial concentrations and reaction rates. Each random set was used to establish a baseline of potential WS expression making it possible to evaluate whether a set of mutations results in a WS type. Effectively, this approach allows sampling of the probability distribution $P (WS | m_i \in Wsp)$ used in our Bayesian model.

We randomly sample 1000 different sets of reaction rates and initial concentrations from uniform priors: reaction rates were sampled randomly from a uniform distribution on log space (i.e. $10^{U[-2,2]}$) and initial concentrations of reactants were sampled from a uniform distribution U[0,10]. For each set, the appropriate differential equation model was integrated and the steady state concentration of the compounds that correspond to a wrinkly spreader (RR in Aws, R* in Wsp and D* for Mws) computed. This served as a baseline for the non-WS phenotype that was used for comparison to determine whether combinations of mutations result in increased WS expression. After obtaining the baseline, we implemented particular combinations of enabling/disabling mutations ($a\ m_i$). Ideally, a distribution linking enabling/disabling mutations to a fold change in reaction rates would be used, but this information is unavailable. In order to progress the effect sizes for enabling and disabling mutations were sampled from $10^{U[0,2]}$ and $10^{U[-2,0]}$, respectively, and then multiplied by the reaction rates. The differential equations were then solved for the same time that it took the baseline to reach steady state. The final concentration of R* (*Figure 4A*), RR (*Figure 4B*) and D* (*Figure 4C*) was then compared to the baseline and the number of times out of 1000 that the WS-inducing compound increased served as an estimate of $P (WS|m_i \in Wsp)$. The probability distribution stabilized by 500 random samples and additional sampling did not produce significant changes (data not shown).

The absence of empirical data on reaction rates, initial concentrations, and expected mutation effect size meant using a random sampling approach requiring estimates for parameter ranges. Parameter ranges were chosen to be broad enough to capture differences spanning several orders of magnitudes while allowing numerical computations for solving the differential equations. To assess the effect of these ranges on the results, the sampling procedure was repeated for WSP for three different parameter regimes (i) an expanded range for initial concentrations [0–50], (ii) an expanded range for reaction rates $10^{[-3,3]}$, (iii) a compressed range for mutational effect size $10^{[-1,1]}$. This analysis shows that qualitative results are robust to these changes (see *Figure 5—figure supplement 3*).

Source code and equations are available in *Supplementary file 1*.

## Additional information

### Competing interests

Paul B Rainey: Reviewing editor, *eLife*. The other authors declare that no competing interests exist.

### Funding

| Funder | Grant reference number | Author |
|---|---|---|
| Royal Society of New Zealand | Marsden Fund Council | Paul B Rainey |

The funders had no role in study design, data collection and interpretation, or the decision to submit the work for publication.

### Author contributions

Peter A Lind, Conceptualization, Data curation, Investigation, Visualization, Methodology, Writing—original draft, Writing—review and editing; Eric Libby, Conceptualization, Software, Formal analysis, Investigation, Visualization, Methodology, Writing—original draft, Writing—review and editing; Jenny Herzog, Data curation, Investigation, Methodology; Paul B Rainey, Conceptualization, Supervision, Funding acquisition, Validation, Visualization, Methodology, Writing—original draft, Project administration, Writing—review and editing

### Author ORCIDs

Peter A Lind (iD) http://orcid.org/0000-0003-1510-8324
Paul B Rainey (iD) http://orcid.org/0000-0003-0879-5795

### Decision letter and Author response

Decision letter https://doi.org/10.7554/eLife.38822.026
Author response https://doi.org/10.7554/eLife.38822.027

## Additional files

### Supplementary files

• Supplementary file 1. Equations and source code.
DOI: https://doi.org/10.7554/eLife.38822.022

• Transparent reporting form
DOI: https://doi.org/10.7554/eLife.38822.023

### Data availability

All data generated or analysed during this study are included in the manuscript and supporting files. Source data files have been provided for Figures 6 and 9.

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

## Appendix 1

DOI: https://doi.org/10.7554/eLife.38822.024

# Predicting mutational routes to new adaptive phenotypes

Peter A. Lind, Eric Libby, Jenny Herzog and Paul B. Rainey

## Analysis of mutations and molecular effects

*Wsp pathway*: Mutations were identified in five genes of the seven-gene pathway all of which were predicted by null model IV (*Figure 5—figure supplement 1*). The most commonly mutated gene was *wspA* (PFLU1219), with ten out of 15 mutations (*Figure 6*) being amino acid substitutions (six unique) clustered in the region 352–420 at the stalk of the signalling domain. This region has been implicated in trimer-of-dimer formation for the WspA homologue in *Pseudomonas aeruginosa* (*O'Connor et al., 2012*) which is critical for self-assembly and localization of Wsp clusters in the membrane. It is possible that these mutations stabilize trimer of dimer formation, change the subcellular location of the Wsp complex, or affect interaction with WspD (putative interface 383–420 in WspA) (*Griswold et al., 2002*) and thus affecting relay of signal to WspE. These effects we interpreted as enabling mutations increasing $r_3$ in *Figure 4A*. The four additional mutations were in frame deletions in a separate region of the transducer domain (ΔT293 - E299, ΔA281-A308). Although it is possible that these mutations could also affect trimer-of-dimer formation, there are predicted methylation sites in the region (*Rice and Dahlquist, 1991*) that regulate the activity of the protein via methyltransferase WspC and methylesterase WspF. Given that disabling mutations are more common than enabling mutations it is likely that these mutations decrease $r_2$ in *Figure 4A* by disrupting the interaction with WspF. We also identified a single mutation that fused the open reading frame of WspC, the methyltransferase that positively regulates WspA activity, to WspD, resulting in a chimeric protein (*Figure 6*, *Figure 6—source data 1*). This mutation is likely to be a rare enabling mutation that increases the activity of WspC (increasing $r_1$ in *Figure 4A*) by physically tethering it to the WspABD complex thus allowing it to more effectively counteract the negative regulator WspF. Alternatively, the tethering may physically block the interaction with WspF (decrease of $r_2$ in *Figure 4A*).

The second most commonly mutated gene in the *wsp* operon was *wspE* (PFLU1223) (*Figure 6*). Four amino acids were repeatedly mutated in the response regulatory domain of WspE and all cluster closely in a structural homology model made with Phyre2 (*Kelley et al., 2015*). All mutated residues surround the active site of the phosphorylated D682 and it is likely that they disrupt feedback regulation by decreasing phosphorylation of the negative regulator WspF (decreasing $r_6$) rather than increasing activation of WspR ($r_5$ in *Figure 4A*).

Twelve mutations were detected in *wspF* (PFLU1224). These are distributed throughout the gene and include amino acid substitutions, in-frame deletions as well as a frame-shift and a stop codon (*Figure 6*). The pattern of mutations is consistent with both the role of WspF as a negative regulator of WspA activity and the well-characterised effect of loss-of-function mutations in this gene (*Bantinaki et al., 2007*; *McDonald et al., 2009*). The mutations are interpreted as decreasing $r_2$ in *Figure 4A*.

Five mutations were found in WspR (PFLU1225), the DGC output response regulator that produces c-di-GMP and activates expression of cellulose (*Figure 6*). All mutations were located in the linker region between the response regulator and DGC domains. Mutations in this region are known to generate constitutively active *wspR* alleles by relieving the requirement for phosphorylation (*Goymer et al., 2006*). They may additionally affect subcellular clustering of WspR (*Huangyutitham et al., 2013*) or shift the equilibrium between the dimeric form of WspR, with low basal activity, towards a tetrameric activated form (*De et al., 2009*). In our model these increase reaction $r_5$.

*Aws pathway*: Mutations were identified in all three genes of the Aws pathway – all of which were predicted by the null Model IV. In the Aws pathway, mutations were most commonly found in *awsX* (25 out of 41 mutations (*Figure 6*). The above-mentioned mutational

hotspot produced in-frame deletions likely mediated by 6 bp direct repeats (*Figure 6—source data 1*). The deletions are consistent with a loss of function and a decrease in $r_3$ (*Figure 4B*) that would leave the partially overlapping open reading frame of the downstream gene (*awsR*) unaffected.

The DGC AwsR, was mutated in 14 cases with an apparent mutational hot spot at T27P (nine mutants) in a predicted transmembrane helix (amino acids 19–41). The remaining mutations were amino acid substitutions in the HAMP linker and in the PAS-like periplasmic domain between the two transmembrane helices. These amino acid substitutions are distant to the output DGC domain (*Figure 6*) and their effects are difficult to interpret, but they could cause changes in dimerization (*Malone et al., 2012*) or the packing of HAMP domains, which could, in turn, alter transmission of conformational changes in the periplasmic PAS-like domain to the DGC domain causing constitutive activation (*Parkinson, 2010*). Such effects would increase $r_4$ in *Figure 4B*. Mutations in the N-terminal part of the protein are easier to interpret based on the existing functional model (*Malone et al., 2012*) and most likely disrupt interactions with the periplasmic negative regulator AwsX resulting in a decrease in $r_3$ in *Figure 4B*.

Two mutations were found in the outer membrane lipoprotein protein AwsO between the signal peptide and the OmpA domain (*Figure 6*). Both mutations were glutamine to proline substitutions (Q34P, Q40P), which together with a previously reported G42V mutation (*McDonald et al., 2009*) suggest that multiple changes in this small region can cause a WS phenotype. This is also supported by data from *Pseudomonas aeruginosa* in which mutations in nine different positions in this region lead to a small colony variant phenotype similar to WS (*Malone et al., 2012*). A functional model based on the YfiBNR in *P. aeruginosa* (*Malone et al., 2012*; *Xu et al., 2016*), suggest that AwsO sequesters AwsX at the outer membrane and that mutations in the N-terminal part of the protein lead to constitutive activation and increased binding of AwsX. This would correspond to an increase in $r_2$ in *Figure 4B*, which would relieve negative regulation of AwsR.

Mws pathway: The MwsR pathway (comprising just a single gene) harbours mutations in both DGC and phosphodiesterase (PDE) domains. Mainly mutations in the C-terminal phosphodiesterase (PDE) domain were predicted (*Figure 4C*). Eleven of 18 mutations were identical in-frame deletions (ΔR1024-E1026) in the PDE domain, mediated by 8 bp direct repeats (*Figure 6*, *Figure 6—source data 1*). It has been shown previously that deletion of the entire PDE domain generates the WS phenotype (*McDonald et al., 2009*), suggesting a negative regulatory role that causes a decrease of $r_2$ in the model in *Figure 4C*. One additional mutation was found in the PDE domain (E1083K) located close to R1024 in a structural homology model made with Phyre2 (*Kelley et al., 2015*), but distant to the active site residues (E1059-L1061). Previously reported mutations (A1018T, ins1089DV) (*McDonald et al., 2009*) are also distant from the active site and cluster in the same region in a structural homology model. This suggests that loss of phosphodiesterase activity may not be the mechanism leading to the WS phenotype. This is also supported by the high solvent accessibility of the mutated residues, which indicates that major stability-disrupting mutations are unlikely and changes in interactions between domains or dimerization are more probable. Thus, it is likely that the WS phenotype resulting from a deletion in the PDE domain is caused by disruption of domain interactions or dimerization rather than loss of phosphodiesterase activity.

The remaining mutations within *mwsR* are amino acid substitutions in the GGDEF domain, close to the DGC active site (G927-F931) with the exception of a duplication of I978-G985. While it is possible that these mutations directly increase the catalytic activity of the DGC, increasing $r_1$ in *Figure 4C*, such enabling mutations are considered to be rare. An alternative hypothesis is that these mutations either interfere with c-di-GMP feedback regulation or produce larger conformational changes that change inter-domain or inter-dimer interactions, similar to the mutations in the PDE domain. Based on these data we reject the current model of Mws function, which predicted mutations decreasing $r_2$ (*Figure 4C*) through mutations inactivating the PDE domain. We instead suggest that the mutations are likely to disrupt the conformational dynamics between the domains and could be seen either as activating

mutations causing constitutive activation or disabling mutations with much reduced mutational target size that must specifically disrupt the interaction surface between the domains. In both cases the previous model leads to an overestimation of the rate to WS for the Mws pathway.

