## [Decision Letter]

[Editors’ note: the authors were asked to provide a plan for revisions before the editors issued a final decision. What follows is the editors’ letter requesting such plan.]

Thank you for submitting your article "Disentangling the effects of genetic architecture, mutational bias and selection on evolutionary forecasting" for consideration by *eLife*. Your article has been reviewed by Patricia Wittkopp as the Senior Editor, a Reviewing Editor, and three reviewers. The following individuals involved in review of your submission have agreed to reveal their identity: James Bull (Reviewer #3).

Overall, we were impressed by the experimental data presented, but had concerns about the predictions and model testing. These concerns included both the clarity of presentation and the substance of the modeling. Different reviewers had different interpretations of some elements of the work because key elements (including Figure 3) were not clearly explained. The mutation model was one element identified as particularly difficult to understand. The lack of statistical tests comparing the predictions to data was also identified as a missing element. Especially in light of the title of the paper; making sure these predictions are made and tested appropriately, is key - we agree that this is where some of the most exciting aspects of the paper lie. Critical points raised during our discussion are provided below. Minimally, we request that you make it clear that you are evaluating two different prediction models and clarify them. Ideally, we'd also like you to make the evolution model explicit and quantitative as well as evaluate each model separately by computing numerical predictions and reporting statistics on the fit with results.

We recognize that the refinement of the modeling work we are requesting may be substantial. Given the list of essential revisions, the editors and reviewers invite you to respond within the next two weeks with an action plan and timetable for the completion of the additional work. We plan to share your responses with the reviewers and then issue a binding recommendation.

This paper reports important experimental work that could easily become a paradigm copied by others. We appreciated the study's span of known biochemical pathways for a phenotype, of documented mutations and mutation counts in those pathways leading to that phenotype, and the attempt to model/quantify the mutation rates in light of some genomic details. The topic is of sufficient interest to publish in *eLife*.

1) A major strength of the study was that there is an in-depth analysis of different mutational pathways to the WS phenotype. This is a data-intensive part of the paper with lots of detail about gene identity of mutations that were found in WS mutants. However, descriptions of gene functions would be better presented in a table with the in text discussion streamlined more.

2) A much better description of Figure 3 is needed. Not only are the colors not explained, but the meaning of the arrows isn't given either (are they quantitative, qualitative; how do they plug into the model?). I thus cannot easily understand how the figure connects to their models. We ultimately found this information in the supplementary materials, but the reader shouldn't need to look there to understand this key element of the paper. As one reviewer wrote: the manuscript describes Figure 3 as a diagram of a "pathway" or "molecular network" but it does not use ordinary conventions for biochemical or regulatory pathways. It is better described as a graph of mathematical dependencies between reaction rates (red) and gene-product concentrations (blue). Once you know this, you can read out some of the implied reactions like F + E* ==(r6)==> E + F*. In the supplement, see reaction 6, Figure 3—figure supplement 1, "Phosphorylated WspE can phosphorylate WspF". It would be more understandable to just list the reactions, I think.

3) Most importantly, we felt that the modeling elements were presented superficially and were not well-integrated. There are two different prediction models developed and evaluated in the paper: one of mutation rates and one of evolutionary tendencies. Regarding mutational predictability, we have (1) a model for predicting relative mutability for 3 pathways, (2) measured mutation rates for the 3 pathways, and (3) a revised model. But there are no statistics telling us how well the model fits and how much it was improved.

In theoretical population genetics there is a simple "origin-fixation" or SSWM framework for considering the relative chances of different discrete evolutionary changes. Whether or not one wishes to criticize this formalism for being too simplistic, it is the obvious framework for making a prediction when one has measured mutation rates and selection coefficients. It allows precise quantitative statements about the relative contributions of selection and mutational heterogeneity to parallel evolution, as per "What drives parallel evolution?" by Bailey et al., 2017, which uses the theory explained more thoroughly in "Parallel evolution: what does it (not) tell us and why is it (still) interesting?" by Lenormand et al., 2016. The analysis also needs statistical tests, e.g., tests of differences where the authors are making claims about differences.

Regarding the modeling framework, let me explain this concretely. The relevant facts on mutation rate, selection coefficient, and frequency evolved are (subsection “Obtaining an unbiased measure of pathway-specific mutation rates to WS”, Figure 6 and Figure 7):

Wsp: u = 3.7E-9; s ~ x; E = 15

Aws: u = 6.5E-9; s ~ x – 0.2; E = 7

Mws: u = 7.4E-10; s ~ x – 0.05; E = 3

Here x is the selection coefficient of the WspF control used in growth competitions, and I have estimated s by eye from Figure 6.

The authors talk about predictability, but they do not compute a prediction using a model, nor compute the fit between results and predictions. One way to do this is to invoke origin-fixation dynamics, yielding 3 relative predictions of u * Pr_fixation(s), which can be compared to the 3 E values using a chi-squared test (or, better, define expected Wsp and Aws as sums over the 5 and 3 relevant gene targets). Given all the numbers and Kimura's formula for Pr_fixation(s, N), this can be done in a spreadsheet in 15 minutes.

If the authors do not want to commit to the assumptions underlying origin-fixation dynamics, they can just do a multinomial regression on the assumption that E(p) = f(u_p, s_p) where p is the pathway.

In summary, we all agreed that under the origin-fixation framework, having estimates of mutation rates for different phenotypes is important and that these are not easy to obtain in most systems. We are excited about the opportunity to test how well a model based on known information can recapitulate observed mutation rates. It is interesting that it does a relatively poor job unless they incorporate information about mutational hotspots. We think that making this analysis more formal in addition to formally testing your evolutionary predictions using an origin-fixation framework (or similar approach) would both strengthen the paper and be in the spirit of what the title claims the paper does.

[Editors’ note: formal revisions were requested, following approval of the authors’ plan of action.]

Thank you for describing your proposed revisions. This summary has been reviewed and discussed by the reviewers and myself. We all agree that your proposed responses to everything except point 3 is satisfactory. For point 3, the reviewers were disappointed in the response. They all agreed that their recommendation to take advantage of your rich and powerful dataset to model the evolutionary process more formally had been dismissed too quickly.

After considering your response and their comments, it seems to me that you are thinking about "predictability" in the way that David Stern does (i.e., which genes/pathways are most likely to be changed to give rise to a particular phenotypic change given the molecular/developmental functions of all genes in the genome), and they are thinking about predictability in a more population/quantitative genetics sense (i.e., given that you know the relative mutation rates and fitness effects for different genes, you should be able to make predictions using theoretical models about how the trait should evolve, which you can then compare to the observed data). In the reviewers’ opinion, you are focused more on the effects of mutations as they arise and they want to see these combined with other evolutionary forces to make more robust evolutionary predictions.

I do not want to be so heavy handed in the review process as to try force you to make your paper something you didn't intend it to be. However, I also see the potential for modeling with your data that has the reviewers excited. It is rare to have the type of data that you have, and there are well-developed models that could make your data even more impactful.

Ultimately, I will leave it up to you to decide whether or not to go beyond the response you propose and fit a specific evolutionary model and compare it to your observed data with statistical tests. If you choose not to do this, however, it is essential that the title, abstract, and other parts of the paper are modified so that readers do not think this type of analysis is included within the paper.

---

## [Author Response]

[Editors’ note: what follows is the authors’ plan to address the revisions.]

Overall, we were impressed by the experimental data presented, but had concerns about the predictions and model testing. These concerns included both the clarity of presentation and the substance of the modeling. Different reviewers had different interpretations of some elements of the work because key elements (including Figure 3) were not clearly explained. The mutation model was one element identified as particularly difficult to understand. The lack of statistical tests comparing the predictions to data was also identified as a missing element. Especially in light of the title of the paper; making sure these predictions are made and tested appropriately, is key – we agree that this is where some of the most exciting aspects of the paper lie. Critical points raised during our discussion are provided below. Minimally, we request that you make it clear that you are evaluating two different prediction models and clarify them. Ideally, we'd also like you to make the evolution model explicit and quantitative as well as evaluate each model separately by computing numerical predictions and reporting statistics on the fit with results.We recognize that the refinement of the modeling work we are requesting may be substantial. Given the list of essential revisions, the editors and reviewers invite you to respond within the next two weeks with an action plan and timetable for the completion of the additional work. We plan to share your responses with the reviewers and then issue a binding recommendation.This paper reports important experimental work that could easily become a paradigm copied by others. We appreciated the study's span of known biochemical pathways for a phenotype, of documented mutations and mutation counts in those pathways leading to that phenotype, and the attempt to model/quantify the mutation rates in light of some genomic details. The topic is of sufficient interest to publish in eLife.1) A major strength of the study was that there is an in-depth analysis of different mutational pathways to the WS phenotype. This is a data-intensive part of the paper with lots of detail about gene identity of mutations that were found in WS mutants. However, descriptions of gene functions would be better presented in a table with the in text discussion streamlined more.

We propose including a table similar to that pasted below into which we move information about the probable molecular effects and the effects of reaction rates on different classes of mutations. This will allow streamlining of the text and improved focus on the motivation for determining the molecular effects on mutations. This is important because it forms the link between targeted genes and reaction rates. It will also allow more attention to be given to the proposed effects of mutations on gene function and reaction rates and their agreement (or not) with the model. However, in removing this information to a table we are concerned that we may leave the reader interested in important detail frustrated. To accommodate this possibility we propose including a supplementary text that complements the table in which we will include relevant detail and where necessary, nuanced interpretations (removed from the current text). This is readily achievable.

**Protein**
**Mutation class**
**Proposed molecular effect**
**Proposed reaction rate effect**
**Reference**
**s**
WspAAmino acid substitutions 352-420trimer-of-dimer formation, localization of Wsp clusters, interaction with WspDIncrease r3O'Connor et al. 2012, Griswold, et al. 2002,WspADeletions A281-A308Disrupt demethylation by WspFDecrease r2Rice and Dahlquist 1991WspC/DFusion of WspC and WspDIncreased methylation by WspC, blocking access for WspFIncrease r1, decrease r2WspEResponse regulatory domain phosphorylation siteDisrupt phosphorylation of WspFDecrease r6Homology modelWspFAny disabling mutationLoss of negative regulation by WspF by demethylation of WspADecrease r2Bantinaki, et al. 2007, McDonald, et al. 2009WspRAmino acid substitutions and small deletion in linker region between response regulator and DGC domainConstitutive activation of DGC without phosphorylation. Effects on subcellular clustering or multimeric state.Increase r5Goymer, et al. 2006,Huang yutitham, et al. 2013, De, et al. 2009

AwsXIn frame loss of function mutationsLoss of negative regulation of AwsRDecrease r3Malone, et al. 2012AwsRAmino acid substitutions transmembrane helix, periplasmic domainDisruption of interaction with AwsXDecrease r3Malone, et al. 2012AwsRAmino acid substitutions HAMP linker domainChanges in dimerization, packing of HAMP domains causing constitutive activationIncrease r4Malone, et al. 2012, Parkinson 2010AwsOAmino acid substitutions in between signal peptide and OmpA domainConstitutive activation and sequestering of AwsXIncrease r2McDonald, et al. 2009, Malone, et al. 2012, Xu, et al. 2016

MwsRAmino acid substitutions and small in frame deletions in interface between EAL and DGC domainsChanges to interdomain interaction or dimerization leading to constitutive activation of DGC functionDecrease r2, increase r1Homology model, McDonald, et al. 2009MwsRAmino acid substitutions near DGC active siteLoss of feedback regulation or changes to interdomain interactionDecrease r2, increase r1Homology model, McDonald, et al. 2009

2) A much better description of Figure 3 is needed. Not only are the colors not explained, but the meaning of the arrows isn't given either (are they quantitative, qualitative; how do they plug into the model?). I thus cannot easily understand how the figure connects to their models. We ultimately found this information in the supplementary materials, but the reader shouldn't need to look there to understand this key element of the paper. As one reviewer wrote: the manuscript describes Figure 3 as a diagram of a "pathway" or "molecular network" but it does not use ordinary conventions for biochemical or regulatory pathways. It is better described as a graph of mathematical dependencies between reaction rates (red) and gene-product concentrations (blue). Once you know this, you can read out some of the implied reactions like F + E* ==(r6)==> E + F*. In the supplement, see reaction 6, Figure 3—figure supplement 1, "Phosphorylated WspE can phosphorylate WspF". It would be more understandable to just list the reactions, I think.

We agree that a better description of Figure 3 is required. In response to your point 3 (below) we propose to include null models for each of the pathways. These will allow a better introduction / motivation and description of our mathematical model that will lead to a revision of Figure 3. We will take care to ensure that the caption serves as a complete description of the figure and its significance. This will require one month.

3) Most importantly, we felt that the modeling elements were presented superficially and were not well-integrated. There are two different prediction models developed and evaluated in the paper: one of mutation rates and one of evolutionary tendencies. Regarding mutational predictability, we have (1) a model for predicting relative mutability for 3 pathways, (2) measured mutation rates for the 3 pathways, and (3) a revised model. But there are no statistics telling us how well the model fits and how much it was improved.In theoretical population genetics there is a simple "origin-fixation" or SSWM framework for considering the relative chances of different discrete evolutionary changes. Whether or not one wishes to criticize this formalism for being too simplistic, it is the obvious framework for making a prediction when one has measured mutation rates and selection coefficients. It allows precise quantitative statements about the relative contributions of selection and mutational heterogeneity to parallel evolution, as per "What drives parallel evolution?" by Bailey et al., 2017, which uses the theory explained more thoroughly in "Parallel evolution: what does it (not) tell us and why is it (still) interesting?" by Lenormand et al., 2016. The analysis also needs statistical tests, e.g., tests of differences where the authors are making claims about differences.

We have given this particular attention. It seems, especially from the suggestion that we borrow from origin-fixation models, there is some misunderstanding of what we have done: our motivation, aims and overall long-term aspiration. Clearly, we need to improve.

Ultimately – although perhaps not for some time – we think that it will be possible to take a given genome sequence and, armed with knowledge of the genotype-to-phenotype map underpinning a particular phenotype, combined with knowledge of localised mutational biases, predict the genetic bases of phenotypic evolution in the absence of additional data. Our work is a small step in this direction and perhaps its most important contribution is to outline a way forward and uniquely (because of our experimental system) assess its validity in terms of what is currently known, what is not known, and what needs (in the future) to be known.

Importantly, our approach aids understanding of *why* evolution follows a particular route. Indeed in our previous work (McDonald et al.,2009; Lind et al.,2015) we have shown the genotype-to-phenotype map to be a significant contributor to parallel evolution (it’s not all about selection and mutation rate). Our work thus contributes to development of a body of knowledge that deepens understanding of the mechanisms of evolution.

Our approach is readily extendable to other organisms that harbour Wsp, Aws and Mws pathways. Precisely this transferability is demonstrated in Lind 2018 (https://www.biorxiv.org/content/early/2018/06/08/342261). By extension to other systems, this stands to feedback to affect the value and predictive power of bottom-up approaches in the future. Further, our approach is amenable to the study of future evolutionary change, for example, that involving reverse evolution of the WS phenotype as evident in studies of lifecycle evolution (Hammerschmidt et al., 2014). In contrast, approaches involving measurement of mutation rate and selection coefficient will be — in absence of mechanistic understanding — specific to each particular instance of adaptive evolution.

We appreciate that origin-fixation models can be useful in predicting the rise and fixation of new mutants. As you and the referees point out such models require estimation of (1) the probability that a mutant will arise and (2) the selection coefficient of that mutant. These parameters, particularly the first, are difficult to predict a priori and usually require fitting to experimental data or an assumption about the probability distribution of mutations. The framework introduced in our paper is different, in that it attempts to estimate the probability that a particular mutant (Wsp, Aws, Mws) will arise in the absence of experimental data (with only knowledge of the genes in a pathway and how they interact to produce a particular phenotype). While we compare predictions arising from our models to experimental data, we do not use these data in our predictions.

Initially we make incorrect predictions based solely on knowledge of the genotype-to-phenotype map (although we are not out by much) but show that we are able to refine predictions through incorporation of information concerning mutational hotspots. The capacity to weigh the influences of gene interactions with gene mutation rates avoids lumping these factors into a single term as is the norm for origin-fixation models. In the revised version of this paper, we propose to spell out these differences more clearly.

In addition to the comments above we believe that using an origin-fixation framework is not appropriate here. Although the selection coefficients could of course be used in combination with mutation rates to predict the probability of fixation this is not what we are trying to do, moreover to do so would be to mislead the reader. There are three reasons. Firstly, the fitness of WS is frequency-dependent and thus a mutant’s selection coefficient is very different at the time it originates compared to when it is established at the air-liquid interface (Rainey and Travisano,1998; Lind et al.,2015). Secondly, the selection coefficient is also dependent on the frequency of other WS types and other phenotypes selected at the air-liquid interface (Lind et al., 2017, Ferguson et al.,2013). Thirdly, fitness in our study is measured against a *wspF* (WS) mutant and not the wild type ancestor, which is needed for use in origin-fixation models. In our experiment, it is not the case that fitness can be estimated by using s=x-0.2 for *aws* as suggested in the comments.

The way forward: we propose as a vehicle for improvement, inclusion of null models for each of the pathways in which we firstly ignore genetic architecture. This means the first line of prediction – as to which pathway evolution will follow – will be based solely on target size. Preliminary analyses show that these null models are hopelessly inaccurate. Next, as we better develop the argument for the importance of genetic architecture, we propose a set of three null models that incorporate genetic architecture with the same number of components and interactions but with randomly assigned architectures. Comparison between these arbitrary models and those informed by current knowledge can then be made – and will be statistically appraised. Together these comparisons will demonstrate the value of incorporating valid architectural information. Finally, as we move to introduce known mutational bias into our models (based on our experimental data) we will compare these to null models where the hotspot is placed randomly in the pathway. Again, we will subject the comparisons to statistical testing. The proposed new analyses will lead to clarifications in the Introduction and Discussion concerning our objectives and the differences with current approaches and particularly the difference between our strategy and origin-fixation models.

This analysis will be completed within two months and will offer many points of comparison and demonstrate the value of the predictive power of our approach.

Additionally, we propose: (1) to perform a statistical analysis of differences between all mutation rates (Figure 2) and fitness (for example *wspA* and *wspR* vs. the control in Figure 6). The number of replicates will be added to all figures. (2) Provide a better motivation for our measurements of fitness: an explanation of why mutations in certain genes have not previously been found (*wspA* and *wspE*) and why Aws is is more rarely used than Wsp despite a higher mutation rate to WS. (3) Add information concerning controls for fitness assays in order to account for effects of the kanamycin marker and double deletions and clarify that the number of mutants per independent culture for the mutation rate assay is <0.5 so that differences in fitness between different WS mutants thus making the point that these have minimal influence on results. These modifications will be completed within a month.

Regarding the modeling framework, let me explain this concretely. The relevant facts on mutation rate, selection coefficient, and frequency evolved are (subsection “Obtaining an unbiased measure of pathway-specific mutation rates to WS”, Figure 6 and Figure 7):Wsp: u = 3.7E-9; s ~ x; E = 15Aws: u = 6.5E-9; s ~ x – 0.2; E = 7Mws: u = 7.4E-10; s ~ x – 0.05; E = 3Here x is the selection coefficient of the WspF control used in growth competitions, and I have estimated s by eye from Figure 6.The authors talk about predictability, but they do not compute a prediction using a model, nor compute the fit between results and predictions. One way to do this is to invoke origin-fixation dynamics, yielding 3 relative predictions of u * Pr_fixation(s), which can be compared to the 3 E values using a chi-squared test (or, better, define expected Wsp and Aws as sums over the 5 and 3 relevant gene targets). Given all the numbers and Kimura's formula for Pr_fixation(s, N), this can be done in a spreadsheet in 15 minutes.If the authors do not want to commit to the assumptions underlying origin-fixation dynamics, they can just do a multinomial regression on the assumption that E(p) = f(u_p, s_p) where p is the pathway.In summary, we all agreed that under the origin-fixation framework, having estimates of mutation rates for different phenotypes is important and that these are not easy to obtain in most systems. We are excited about the opportunity to test how well a model based on known information can recapitulate observed mutation rates. It is interesting that it does a relatively poor job unless they incorporate information about mutational hotspots. We think that making this analysis more formal in addition to formally testing your evolutionary predictions using an origin-fixation framework (or similar approach) would both strengthen the paper and be in the spirit of what the title claims the paper does.

[Editors’ notes: the authors’ response after being formally invited to submit a revised submission follows.]

Thank you for describing your proposed revisions. This summary has been reviewed and discussed by the reviewers and myself. We all agree that your proposed responses to everything except point 3 is satisfactory. For point 3, the reviewers were disappointed in the response. They all agreed that their recommendation to take advantage of your rich and powerful dataset to model the evolutionary process more formally had been dismissed too quickly.After considering your response and their comments, it seems to me that you are thinking about "predictability" in the way that David Stern does (i.e., which genes/pathways are most likely to be changed to give rise to a particular phenotypic change given the molecular/developmental functions of all genes in the genome), and they are thinking about predictability in a more population/quantitative genetics sense (i.e., given that you know the relative mutation rates and fitness effects for different genes, you should be able to make predictions using theoretical models about how the trait should evolve, which you can then compare to the observed data). In the reviewers’ opinion, you are focused more on the effects of mutations as they arise and they want to see these combined with other evolutionary forces to make more robust evolutionary predictions.I do not want to be so heavy handed in the review process as to try force you to make your paper something you didn't intend it to be. However, I also see the potential for modeling with your data that has the reviewers excited. It is rare to have the type of data that you have, and there are well-developed models that could make your data even more impactful.Ultimately, I will leave it up to you to decide whether or not to go beyond the response you propose and fit a specific evolutionary model and compare it to your observed data with statistical tests. If you choose not to do this, however, it is essential that the title, abstract, and other parts of the paper are modified so that readers do not think this type of analysis is included within the paper.

We have changed the title to “Predicting mutational routes to new adaptive phenotypes”. This we have done after realizing that our previous title gave the unintended message that we were combining mutation and selection into our model of evolutionary forecasting. This, as the referees have correctly understood, is not what we have done. What we have done (by virtue of being able to obtain a set of mutants with and without selection) is to show that there really is a need to link mutations to their fitness effects. We have made sure that this message is consistently made in the Abstract and elsewhere. In the revised Discussion section we emphasize that construction of models that link mutations to their fitness effects is the yet to be achieved Holy Grail of forecasting.

We have added new material to the Introduction to acknowledge origin-fixation models. We point out briefly their limitations, but importantly, we make clear that what we aspire to is quite different. Our hope is that at some future time it will be possible to read a genome sequence and be able to forecast future evolutionary change based on mechanistic understanding of rules underpinning expression of the phenotype from genotype, knowledge of mutational biases and understanding of the connection between mutations and their fitness effects, just as today we use the genome to interrogate various databases to infer much about gene function and regulation.

We have removed from the text the detailed descriptions of the likely effect of mutations on reaction rates of each pathway – leaving just a short summary – and placed this in Table 1.

We have substantially revised the modeling section as per our proposed plan. Specifically, in order to show the thinking behind our modeling approach we have included three new models, that begin with a simple (and intentionally naïve) expression of the relationship between genotype and the probability of generating WS phenotypes and conclude with a comprehensive model that expresses the relationship between genotype and WS phenotype based on interaction rates among pathway components (the model previously presented). We describe this as a graph of mathematical dependencies and provide full details of the models in terms of functional descriptions and system of differential equations in the child figures (Figure 4—figure supplement 1, Figure 4—figure supplement 2 and Figure 4—figure supplement 3). We have also clarified the use of multiple changes to reaction rates by single mutations and modified the model to disregard multiple changes that are likely to require several mutations. This has led to minor changes in the ratios of the different pathways and in also in the Figure 4 (now Figure 5), but they do not change our conclusions.

We have added a new Figure 3 showing the predictions of the different null models compared to the experimental data.

The complex graph depicting interactions among pathway components (previously Figure 3), is now Figure 4 and we have taken care to explain its significance fully both in the text and in the caption.

A new Figure 7 has been added that presents an analysis of the results of the different models the experimentally supported mutational hotspot and the predicted relationship between mutation and likelihood of generating WS via each of the three focal pathways. For the two comprehensive models we include the range of predicted states arising from extremes of all enabling and all disabling reactions allowing the reader to assess the full range of predictions and not just values from a single hotspot as previously. This was previously Figure 4D,E and part of the Figure 4—figure supplement 2B.

We have reorganized the end of the Results section to make it clear to the reader that we do not attempt to predict the outcome of experimental evolution using mutation rates and selection coefficients. Instead we now first present the differences in mutational spectrum with and without selection. We have provided a better motivation for the fitness assays, which is to provide an explanation for these differences. We show that there are significant differences in fitness between different mutants and that the direction of these fitness differences explains the underrepresentation of certain types of mutations in the experimental evolution data thus highlighting the importance of incorporating selection in forecasting evolution. We deal with the origin-fixation framework in the Introduction (see above) touching on various assumptions, in particular the need for interactions between mutants to show no frequency dependence. As our decades long experience with the WS system has shown that it is replete with frequency dependent interactions we make no further mention in the Results section. To attempt to place our data in context of origin-fixation models would constitute misuse of the fitness data.

We have included statistical tests for all claims of differences as well as number of replicates. A table of key resources has been added to the Results section. Information about controls for fitness assays have been added to the methods section and we have clarified that the number of mutants per independent culture is so low that differences in fitness between different WS mutants will have minimal influence on results.

Finally, we have substantially revised the Discussion section with much clearer focus on the issues surrounding mechanistic (bottom-up) evolutionary forecasting.